# DUAL-BALANCING FOR MULTI-TASK LEARNING

## ABSTRACT

Multi-task learning (MTL), a learning paradigm to learn multiple related tasks simultaneously, has achieved great success in various fields. However, task balancing problem remains a significant challenge in MTL, with the disparity in loss/gradient scales often leading to performance compromises. In this paper, we propose a Dual-Balancing Multi-Task Learning (DB-MTL) method to alleviate the task balancing problem from both loss and gradient perspectives. Specifically, DB-MTL ensures loss-scale balancing by performing a logarithm transformation on each task loss, and guarantees gradient-magnitude balancing via normalizing all task gradients to the same magnitude as the maximum gradient norm. Extensive experiments conducted on several benchmark datasets consistently demonstrate the state-of-the-art performance of DB-MTL.

## 1 INTRODUCTION

Multi-task learning (MTL) (Caruana, 1997; Zhang & Yang, 2022) jointly learns multiple related tasks using a single model, improving parameter-efficiency and inference speed compared with learning a separate model for each task. By sharing the model, MTL can extract common knowledge to improve each task's performance. MTL has demonstrated its superiority in various fields, such as computer vision (Liu et al., 2019a; Vandenhende et al., 2020; 2021; Xu et al., 2022; Ye & Xu, 2022), natural language processing (Chen et al., 2021; Liu et al., 2017; 2019b; Sun et al., 2020; Wang et al., 2021), and recommendation systems (Hazimeh et al., 2021; Ma et al., 2018a;b; Tang et al., 2020; Wang et al., 2023).

To learn multiple tasks simultaneously, equal weighting (EW) (Zhang & Yang, 2022) is a straightforward method that minimizes the sum of task losses with equal task weights but usually causes the challenging *task balancing problem* (Lin et al., 2022; Vandenhende et al., 2021), where some tasks perform well but the others perform unsatisfactorily (Standley et al., 2020). A number of methods have been recently proposed to alleviate this problem by dynamically tune the task weights. They can be roughly categorized into *loss balancing* methods (Kendall et al., 2018; Liu et al., 2021b; 2019a; Ye et al., 2021) such as balancing tasks based on learning speed (Liu et al., 2019a) or validation performance (Ye et al., 2021) *at the loss level*, and *gradient balancing* approaches (Chen et al., 2018b; 2020; Fernando et al., 2023; Liu et al., 2021a;b; Navon et al., 2022; Sener & Koltun, 2018; Wang et al., 2021; Yu et al., 2020) such as balancing gradients by mitigating gradient conflicts (Yu et al., 2020) or enforcing gradient norms close (Chen et al., 2018b) *at the gradient level*. Recently, Kurin et al. (2022); Lin et al. (2022); Xin et al. (2022) conduct extensive empirical studies and demonstrate that the performance of existing methods are undesirable, which indicates the task balancing issue is still an open problem in MTL.

In this paper, we focus on simultaneously balancing loss scales at the loss level and gradient magnitudes at the gradient level to mitigate the task balancing problem. Since the loss scales/gradient magnitudes among tasks can be different, a large one can dominate the update direction of the model, causing unsatisfactory performance on some tasks (Liu et al., 2021b; Standley et al., 2020). Therefore, we propose a simple yet effective Dual-Balancing Multi-Task Learning (**DB-MTL**) method that consists of loss-scale and gradient-magnitude balancing approaches. First, we perform a logarithm transformation on each task loss to ensure all task losses have the same scale, which is non-parametric and can recover the loss transformation in IMTL-L (Liu et al., 2021b). We find that the logarithm transformation also benefits the existing gradient balancing methods, as shown in Figure 1. Second, we normalize all task gradients to the same magnitude as the maximum gradient norm, which is training-free and guarantees all gradients' magnitude are the same compared with

GradNorm (Chen et al., 2018b). We empirically find that the magnitude of normalized gradients plays an important role in performance and setting it as the maximum gradient norm among tasks performs the best, as shown in Figure 4, thus it is adopted. We perform extensive experiments on several benchmark datasets and the results consistently demonstrate DB-MTL achieves state-of-the-art performance.

Our contributions are summarized as follows: (i) We propose the DB-MTL method, a dual-balancing approach to alleviate the task-balancing problem, consisting of loss-scale and gradient-magnitude balancing methods; (ii) We conduct extensive experiments to demonstrate that DB-MTL achieves state-of-the-art performance on several benchmark datasets; (iii) Experimental results show that the loss-scale balancing method is beneficial to existing gradient balancing methods.

## 2 RELATED WORKS

Given $T$ tasks and each task $t$ has a training dataset $\mathcal{D}_t$, MTL aims to learn a model on $\{\mathcal{D}_t\}_{t=1}^T$. The parameters of an MTL model consists of two parts: task-sharing parameter $\boldsymbol{\theta}$ and task-specific parameters $\{\boldsymbol{\psi}_t\}_{t=1}^T$. For example, in computer vision tasks, $\boldsymbol{\theta}$ usually represents a feature encoder (e.g., *ResNet* (He et al., 2016)) to extract common features among tasks and $\boldsymbol{\psi}_t$ denotes a task-specific output module (e.g., a fully-connected layer). For parameter efficiency, $\boldsymbol{\theta}$ contains most of the MTL model parameters, which is crucial to the performance.

Let $\ell_t(\mathcal{D}_t; \boldsymbol{\theta}, \boldsymbol{\psi}_t)$ denote the average loss on $\mathcal{D}_t$ for task $t$ using $(\boldsymbol{\theta}, \boldsymbol{\psi}_t)$. The objective function of MTL is $\sum_{t=1}^T \gamma_t \ell_t(\mathcal{D}_t; \boldsymbol{\theta}, \boldsymbol{\psi}_t)$, where $\gamma_t$ is the task weight for task $t$. Equal weighting (EW) (Zhang & Yang, 2022) is a simple approach in MTL by setting $\gamma_t = 1$ for all tasks. However, EW usually causes the task balancing problem where some tasks perform unsatisfactorily (Standley et al., 2020). Hence, many MTL methods are proposed to improve the performance of EW by dynamically tunning task weights $\{\gamma_t\}_{t=1}^T$ during the training process, which can be categorized into loss balancing, gradient balancing, and hybrid balancing approaches.

**Loss Balancing Methods.** This type of method aims at weighting task losses with $\{\gamma_t\}_{i=1}^T$ computed dynamically according to different measures such as homoscedastic uncertainty (Kendall et al., 2018), learning speed (Liu et al., 2019a), validation performance (Ye et al., 2021), and improvable gap (Dai et al., 2023). Different from these methods, IMTL-L (Liu et al., 2021b) expects the weighted losses $\{\gamma_t \ell_t(\mathcal{D}_t; \boldsymbol{\theta}, \boldsymbol{\psi}_t)\}_{t=1}^T$ to be constant for all tasks and performs a transformation on each loss as $e^{s_t} \ell_t(\mathcal{D}_t; \boldsymbol{\theta}, \boldsymbol{\psi}_t) - s_t$, where $s_t$ is a learnable parameter for $t$-th task and approximately solved by gradient descent at every iteration. For loss balancing methods, the task weights $\gamma_t$ affects the update of both task-sharing parameter $\boldsymbol{\theta}$ and task-specific parameter $\boldsymbol{\psi}$.

**Gradient Balancing Methods.** From the gradient perspective, the update of task-sharing parameters $\boldsymbol{\theta}$ depends on all task gradients $\{\nabla_{\boldsymbol{\theta}} \ell_t(\mathcal{D}_t; \boldsymbol{\theta}, \boldsymbol{\psi}_t)\}_{t=1}^T$. Thus, gradient balancing methods aim to aggregate all task gradients in different manners. For example, MGDA (Sener & Koltun, 2018) formulates MTL as a multi-objective optimization problem and selects the aggregated gradient with the minimum norm as in Désidéri (2012). CAGrad (Liu et al., 2021a) improves MGDA by constraining the aggregated gradient to around the average gradient, while MoCo (Fernando et al., 2023) mitigates the bias in MGDA by introducing a momentum-like gradient estimate and a regularization term. GradNorm (Chen et al., 2018b) learns task weights to scale task gradients such that they have close magnitudes. PCGrad (Yu et al., 2020) projects the gradient of one task onto the normal plane of the other if their gradients conflict while GradVac (Wang et al., 2021) aligns the gradients regardless of whether the gradient conflicts or not. GradDrop (Chen et al., 2020) randomly masks out the gradient values with inconsistent signs. IMTL-G (Liu et al., 2021b) learns task weights to enforce that the aggregated gradient has equal projections onto each task gradient. Nash-MTL (Navon et al., 2022) formulates gradient aggregation as a Nash bargaining game. For most gradient balancing methods (Chen et al., 2020; Fernando et al., 2023; Liu et al., 2021a;b; Yu et al., 2020), the task weight $\gamma_t$ only affects the update of task-sharing parameter $\boldsymbol{\theta}$. While in some gradient balancing methods (Chen et al., 2018b; Navon et al., 2022; Sener & Koltun, 2018), task weight $\gamma_t$ acts as the same in loss balancing methods.

**Hybrid Balancing Methods.** As the loss balancing and gradient balancing methods are complementary, these two types of methods can be combined as hybrid balancing methods to achieve better performance. In hybrid balancing methods, the task weight $\gamma_t$ is the product of loss and gradient

balancing weights. IMTL (Liu et al., 2021b), combining IMTL-L with IMTL-G, is the first hybrid balancing method. Dai et al. (2023); Lin et al. (2022); Liu et al. (2022) empirically demonstrate their methods can be combined with some existing loss/gradient balancing methods for further performance improvement. Following this, we propose DB-MTL by combining logarithm transformation (for balancing losses) and the proposed max-norm gradient normalization method (for balancing gradients).

## 3 PROPOSED METHOD

In this section, we alleviate the task balancing problem from both the loss and gradient perspectives. First, we balance all loss scales by performing a logarithm transformation on each task's loss (Section 3.1). Next, we achieve gradient-magnitude balancing via normalizing each task's gradient to the same magnitude as the maximum gradient norm (Section 3.2). The procedure, which will be called DB-MTL (Dual-Balancing Multi-Task Learning), is shown in Algorithm 1.

### 3.1 SCALE-BALANCING LOSS TRANSFORMATION

Tasks with different types of loss functions usually have different loss scales, leading to the task balancing problem. For example, in the *NYUv2* dataset (Silberman et al., 2012), the cross-entropy loss, $L_1$ loss, and cosine loss are used as the loss functions of the semantic segmentation, depth estimation, and surface normal prediction tasks, respectively. As observed in Navon et al. (2022); Standley et al. (2020); Yu et al. (2020) and Table 1 in our experimental results, MTL methods like EW perform undesirably on the surface normal prediction task. For example, in the *NYUv2* dataset, surface normal prediction is dominated by the other two tasks (semantic segmentation and depth estimation), causing undesirable performance.

When prior knowledge of the loss scales is available, we can choose $\{s_t^\star\}_{t=1}^T$ such that $\{s_t^\star \ell_t(\mathcal{D}_t; \boldsymbol{\theta}, \boldsymbol{\psi}_t)\}_{t=1}^T$ have the same scale, and then minimize $\sum_{t=1}^T s_t^\star \ell_t(\mathcal{D}_t; \boldsymbol{\theta}, \boldsymbol{\psi}_t)$. Previous methods (Kendall et al., 2018; Liu et al., 2021b; 2019a; Ye et al., 2021) implicitly learn $\{s_t^\star\}_{t=1}^T$ when learning the task weights $\{\gamma_t\}_{t=1}^T$. However, since the optimal $\{s_t^\star\}_{t=1}^T$ cannot be obtained during training, this can lead to sub-optimal performance.

Logarithm transformation (Eigen et al., 2014; Girshick, 2015) can be used to achieve the same scale for all losses without the availability of $\{s_t^\star\}_{t=1}^T$. Specifically, since $\nabla_{\boldsymbol{\theta}, \boldsymbol{\psi}_t} \log \ell_t(\mathcal{D}_t; \boldsymbol{\theta}, \boldsymbol{\psi}_t) = \frac{\nabla_{\boldsymbol{\theta}, \boldsymbol{\psi}_t} \ell_t(\mathcal{D}_t; \boldsymbol{\theta}, \boldsymbol{\psi}_t)}{\ell_t(\mathcal{D}_t; \boldsymbol{\theta}, \boldsymbol{\psi}_t)}$, it is equivalent to taking gradient of the scaled task loss $\frac{\ell_t(\mathcal{D}_t; \boldsymbol{\theta}, \boldsymbol{\psi}_t)}{\text{stop\_gradient}(\ell_t(\mathcal{D}_t; \boldsymbol{\theta}, \boldsymbol{\psi}_t))}$, where $\text{stop\_gradient}(\cdot)$ is the stop-gradient operation. Note that $\frac{\ell_t(\mathcal{D}_t; \boldsymbol{\theta}, \boldsymbol{\psi}_t)}{\text{stop\_gradient}(\ell_t(\mathcal{D}_t; \boldsymbol{\theta}, \boldsymbol{\psi}_t))}$ has the same scale (i.e., 1) for all tasks.

**Discussion.** Although the logarithm transformation is a simple technique to achieve scale-balancing (Eigen et al., 2014; Girshick, 2015), in MTL, it has been studied in Navon et al. (2022) as one of the baselines. Recently, Dai et al. (2023) propose a loss balancing method using the logarithm transformation, but lack an explanation on why the logarithm transformation works. In this paper, we thoroughly study it and show that logarithm transformation can address the loss scale problem in MTL. Moreover, different from Dai et al. (2023), which studies combining logarithm transformation with existing *loss* balancing methods (e.g., (Lin et al., 2022; Liu et al., 2019a)), we empirically

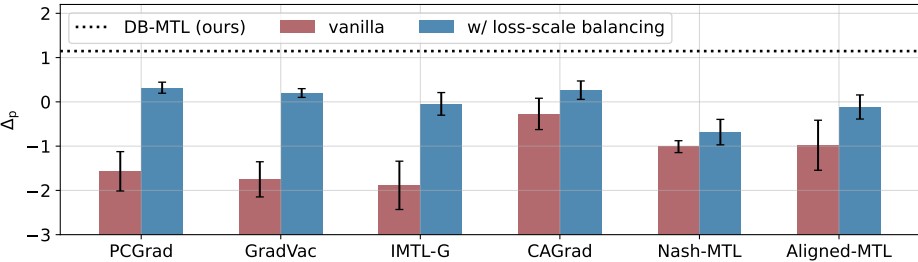

Figure 1: Performance of existing gradient balancing methods with the loss-scale balancing method (i.e., logarithm transformation) on *NYUv2*. "vanilla" stands for the original method.

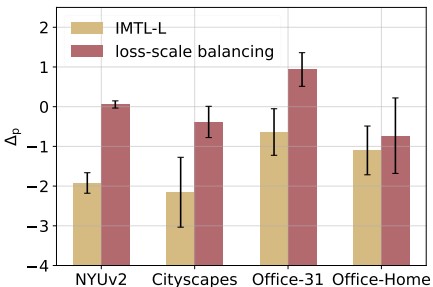
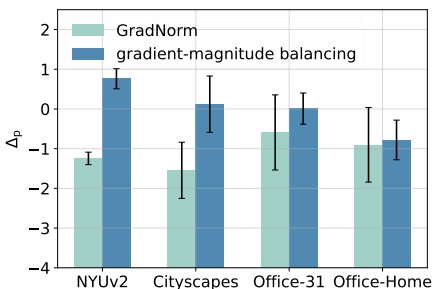

Figure 2: Comparison of IMTL-L (Liu et al., 2021b) and the loss-scale balancing method on four datasets.

Figure 3: Comparison of GradNorm (Chen et al., 2018b) and the gradient-magnitude balancing method on four datasets.

demonstrate that integrating logarithm transformation into existing *gradient* balancing methods (PC-Grad (Yu et al., 2020), GradVac (Wang et al., 2021), IMTL-G (Liu et al., 2021b), CAGrad (Liu et al., 2021a), Nash-MTL (Navon et al., 2022), and Aligned-MTL (Senushkin et al., 2023)) can improve their performance by a large margin, as shown in Figure 1.

IMTL-L (Liu et al., 2021b) tackles the loss scale issue using a transformed loss $e^{s_t}\ell_t(\mathcal{D}_t; \boldsymbol{\theta}, \boldsymbol{\psi}_t) - s_t$, where $s_t$ is a learnable parameter for $t$-th task and approximately solved by one-step gradient descent at every iteration. Hence, it cannot ensure all loss scales are the same in each iteration, while the logarithm transformation can. Proposition A.1 in Appendix shows the logarithm transformation is equivalent to IMTL-L when $s_t$ is the exact minimizer in each iteration. Empirically, Figure 2 shows that logarithm transformation consistently outperforms IMTL-L on four datasets (*NYUv2*, *Cityscapes*, *Office-31*, and *Office-Home*) in terms of $\Delta_{\mathrm{p}}$ (Eq. (2)).

## 3.2 MAGNITUDE-BALANCING GRADIENT NORMALIZATION

In addition to the scale issue in task losses, task gradients also suffer from the scale issue. The update direction by uniformly averaging all task gradients may be dominated by the large task gradients, causing sub-optimal performance (Liu et al., 2021a; Yu et al., 2020).

A simple approach is to normalize task gradients into the same magnitude. For the task gradients, as computing the batch gradient $\nabla_{\boldsymbol{\theta}} \log \ell_t(\mathcal{D}_t; \boldsymbol{\theta}, \boldsymbol{\psi}_t)$ is computationally expensive, a mini-batch stochastic gradient descent method is always used in practice. Specifically, at iteration $k$, we sample a mini-batch $\mathcal{B}_{t,k}$ from $\mathcal{D}_t$ for the $t$-th task ($t = 1, \ldots, T$) (step 5 in Algorithm 1) and compute the mini-batch gradient $\boldsymbol{g}_{t,k} = \nabla_{\boldsymbol{\theta}_k} \log \ell_t(\mathcal{B}_{t,k}; \boldsymbol{\theta}_k, \boldsymbol{\psi}_{t,k})$ (step 6 in Algorithm 1). Exponential moving average (EMA), which is popularly used in adaptive gradient methods (e.g., RMSProp (Tieleman & Hinton, 2012), AdaDelta (Zeiler, 2012), and Adam (Kingma & Ba, 2015)), is used to estimate $\mathbb{E}_{\mathcal{B}_{t,k} \sim \mathcal{D}_t} \nabla_{\boldsymbol{\theta}_k} \log \ell_t(\mathcal{B}_{t,k}; \boldsymbol{\theta}_k, \boldsymbol{\psi}_{t,k})$ dynamically (step 7 in Algorithm 1) as

$$\hat{\boldsymbol{g}}_{t,k} = \beta \hat{\boldsymbol{g}}_{t,k-1} + (1 - \beta) \boldsymbol{g}_{t,k},$$

where $\beta \in (0, 1)$ controls the forgetting rate. After obtaining the task gradients $\{\hat{\boldsymbol{g}}_{t,k}\}_{t=1}^{T}$, we normalize them to have the same magnitude $\ell_2$ norm, and compute the aggregated gradient as

$$\tilde{\boldsymbol{g}}_k = \alpha_k \sum_{t=1}^{T} \frac{\hat{\boldsymbol{g}}_{t,k}}{\|\hat{\boldsymbol{g}}_{t,k}\|_2}, \tag{1}$$

where $\alpha_k$ is a scale factor controlling the update magnitude. After normalization, all tasks contribute equally to the update direction.

The choice of $\alpha_k$ is critical for alleviating the task balancing problem. Intuitively, when some tasks have large gradient norms and others have small gradient norms, it means the model $\boldsymbol{\theta}_k$ is close to a point where the former tasks have not yet converged while the latter tasks have converged. Such point is unsatisfactory in MTL and can cause the task balancing problem since we expect all tasks to achieve convergence. Hence, $\alpha_k$ should be large enough to escape the unsatisfactory point. When all task gradient norms are small, it indicates the model $\boldsymbol{\theta}_k$ is close to a stationary point of all tasks, $\alpha_k$ should be small such that the model will be caught by such point. Thus, we can choose

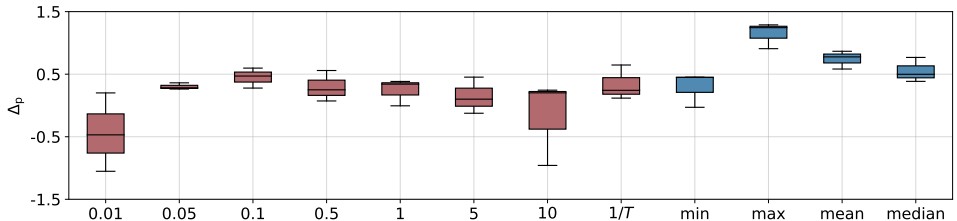

Figure 4: Results of different strategies for $\alpha_k$ in Eq. (1) on the *NYUv2* dataset in terms of $\Delta_{\mathrm{p}}$. "min", "max", "mean", and "median" denotes the minimum, maximum, average, and median of $\|\hat{\boldsymbol{g}}_{t,k}\|_2$ $(t = 1, \ldots, T)$, respectively.

$\alpha_k = \max_{1 \leq t \leq T} \|\hat{\boldsymbol{g}}_{t,k}\|_2$. Note that $\alpha_k$ is small if and only if the norms of all the task gradients are small. Figure 4 shows the performance of using different strategies for adjusting $\alpha_k$ on the *NYUv2* dataset, where the experimental setup is in Section 4.1. Results on the other datasets are in Appendix (Figure 6). As can be seen, the maximum-norm strategy consistently performs much better, and thus it is used (in step 9 of Algorithm 1).

After scaling the losses and gradients, the task-sharing parameter is updated as $\boldsymbol{\theta}_{k+1} = \boldsymbol{\theta}_k - \eta \tilde{\boldsymbol{g}}_k$ (step 10), where $\eta > 0$ is the learning rate. For the task-specific parameters $\{\boldsymbol{\psi}_{t,k}\}_{t=1}^T$, as the update of each of them only depends on the corresponding task gradient separately, their gradients do not have the gradient scaling issue. Hence, the update for task-specific parameters is $\boldsymbol{\psi}_{t,k+1} = \boldsymbol{\psi}_{t,k} - \eta \nabla_{\boldsymbol{\psi}_{t,k}} \log \ell_t(\mathcal{B}_{t,k}; \boldsymbol{\theta}_k, \boldsymbol{\psi}_{t,k})$ (steps 11-13).

**Discussion.** GradNorm (Chen et al., 2018b) aims to learn $\{\gamma_t\}_{t=1}^T$ so that the scaled gradients have similar norms. However, it has two problems. First, alternating the updates of model parameters and task weights cannot guarantee all task gradients to have the same magnitude in each iteration. Second, from Figures 4 and 6, choice of the update magnitude $\alpha_k$ can significantly affect performance, but this is not considered in GradNorm. Figure 3 shows the performance comparison between Grad-Norm and the proposed gradient-magnitude balancing method on four datasets (*NYUv2*, *Cityscapes*, *Office-31*, and *Office-Home*). As can be seen, the proposed method achieves better performance than GradNorm in terms of $\Delta_{\mathrm{p}}$ on all datasets.

---

**Algorithm 1** Dual-Balancing Multi-Task Learning.

---

**Require:** numbers of iterations $K$, learning rate $\eta$, tasks $\{\mathcal{D}_t\}_{t=1}^T$, $\beta$, $\epsilon = 10^{-8}$;
1: randomly initialize $\boldsymbol{\theta}_0, \{\boldsymbol{\psi}_{t,0}\}_{t=1}^T$;
2: initialize $\hat{\boldsymbol{g}}_{t,-1} = \boldsymbol{0}$, for all $t$;
3: **for** $k = 0, \ldots, K - 1$ **do**
4:     **for** $t = 1, \ldots, T$ **do**
5:         sample a mini-batch dataset $\mathcal{B}_{t,k}$ from $\mathcal{D}_t$;
6:         $\boldsymbol{g}_{t,k} = \nabla_{\boldsymbol{\theta}_k} \log(\ell_t(\mathcal{B}_{t,k}; \boldsymbol{\theta}_k, \boldsymbol{\psi}_{t,k}) + \epsilon)$;
7:         compute $\hat{\boldsymbol{g}}_{t,k} = \beta \hat{\boldsymbol{g}}_{t,k-1} + (1 - \beta) \boldsymbol{g}_{t,k}$;
8:     **end for**
9:     compute $\tilde{\boldsymbol{g}}_k = \alpha_k \sum_{t=1}^T \frac{\hat{\boldsymbol{g}}_{t,k}}{\|\hat{\boldsymbol{g}}_{t,k}\|_2 + \epsilon}$, where $\alpha_k = \max_{1 \leq t \leq T} \|\hat{\boldsymbol{g}}_{t,k}\|_2$;
10:    update task-sharing parameter by $\boldsymbol{\theta}_{k+1} = \boldsymbol{\theta}_k - \eta \tilde{\boldsymbol{g}}_k$;
11:    **for** $t = 1, \ldots, T$ **do**
12:        $\boldsymbol{\psi}_{t,k+1} = \boldsymbol{\psi}_{t,k} - \eta \nabla_{\boldsymbol{\psi}_{t,k}} \log(\ell_t(\mathcal{B}_{t,k}; \boldsymbol{\theta}_k, \boldsymbol{\psi}_{t,k}) + \epsilon)$;
13:    **end for**
14: **end for**
15: **return** $\boldsymbol{\theta}_K, \{\boldsymbol{\psi}_{t,K}\}_{t=1}^T$.

---

## 4 EXPERIMENTS

In this section, we empirically evaluate the proposed DB-MTL on a number of tasks, including scene understanding (Section 4.1), image classification (Section 4.2), and molecular property prediction (Section 4.3).

Table 1: Performance on *NYUv2* with 3 tasks. $\uparrow$ ($\downarrow$) means the higher (lower) the result, the better the performance. The best and second best results are marked in **bold** and underline, respectively.

| | Segmentation | | Depth Estimation | | Surface Normal Prediction | | | | | |
| --- | --- | --- | --- | --- | --- | --- | --- | --- | --- | --- |
| | | | | | Angle Distance | | Within $t°$ | | | $\Delta_{\mathrm{p}}\uparrow$ |
| | mIoU$\uparrow$ | PAcc$\uparrow$ | AErr$\downarrow$ | RErr$\downarrow$ | Mean$\downarrow$ | MED$\downarrow$ | 11.25$\uparrow$ | 22.5$\uparrow$ | 30$\uparrow$ | |
| STL | 53.50 | 75.39 | 0.3926 | 0.1605 | 21.99 | **15.16** | 39.04 | **65.00** | 75.16 | 0.00 |
| EW | 53.93 | 75.53 | 0.3825 | 0.1577 | 23.57 | 17.01 | 35.04 | 60.99 | 72.05 | $-1.78_{\pm0.45}$ |
| GLS | 54.59 | **76.06** | 0.3785 | **0.1555** | 22.71 | 16.07 | 36.89 | 63.11 | 73.81 | $+0.30_{\pm0.30}$ |
| RLW | 54.04 | 75.58 | 0.3827 | 0.1588 | 23.07 | 16.49 | 36.12 | 62.08 | 72.94 | $-1.10_{\pm0.40}$ |
| UW | 54.29 | 75.64 | 0.3815 | 0.1583 | 23.48 | 16.92 | 35.26 | 61.17 | 72.21 | $-1.52_{\pm0.39}$ |
| DWA | 54.06 | 75.64 | 0.3820 | 0.1564 | 23.70 | 17.11 | 34.90 | 60.74 | 71.81 | $-1.71_{\pm0.25}$ |
| IMTL-L | 53.89 | 75.54 | 0.3834 | 0.1591 | 23.54 | 16.98 | 35.09 | 61.06 | 72.12 | $-1.92_{\pm0.25}$ |
| IGBv2 | **54.61** | 76.00 | 0.3817 | 0.1576 | 22.68 | 15.98 | 37.14 | 63.25 | 73.87 | $+0.05_{\pm0.29}$ |
| MGDA | 53.52 | 74.76 | 0.3852 | 0.1566 | 22.74 | 16.00 | 37.12 | 63.22 | 73.84 | $-0.64_{\pm0.25}$ |
| GradNorm | 53.91 | 75.38 | 0.3842 | 0.1571 | 23.17 | 16.62 | 35.80 | 61.90 | 72.84 | $-1.24_{\pm0.15}$ |
| PCGrad | 53.94 | 75.62 | 0.3804 | 0.1578 | 23.52 | 16.93 | 35.19 | 61.17 | 72.19 | $-1.57_{\pm0.44}$ |
| GradDrop | 53.73 | 75.54 | 0.3837 | 0.1580 | 23.54 | 16.96 | 35.17 | 61.06 | 72.07 | $-1.85_{\pm0.39}$ |
| GradVac | 54.21 | 75.67 | 0.3859 | 0.1583 | 23.58 | 16.91 | 35.34 | 61.15 | 72.10 | $-1.75_{\pm0.39}$ |
| IMTL-G | 53.01 | 75.04 | 0.3888 | 0.1603 | 23.08 | 16.43 | 36.24 | 62.23 | 73.06 | $-1.89_{\pm0.54}$ |
| CAGrad | 53.97 | 75.54 | 0.3885 | 0.1588 | 22.47 | 15.71 | 37.77 | 63.82 | 74.30 | $-0.27_{\pm0.35}$ |
| MTAdam | 52.67 | 74.86 | 0.3873 | 0.1583 | 23.26 | 16.55 | 36.00 | 61.92 | 72.74 | $-1.97_{\pm0.23}$ |
| Nash-MTL | 53.41 | 74.95 | 0.3867 | 0.1612 | 22.57 | 15.94 | 37.30 | 63.40 | 74.09 | $-1.01_{\pm0.13}$ |
| MetaBalance | 53.92 | 75.57 | 0.3901 | 0.1594 | 22.85 | 16.16 | 36.72 | 62.91 | 73.62 | $-1.06_{\pm0.17}$ |
| MoCo | 52.25 | 74.56 | 0.3920 | 0.1622 | 22.82 | 16.24 | 36.58 | 62.72 | 73.49 | $-2.25_{\pm0.51}$ |
| Aligned-MTL | 52.94 | 75.00 | 0.3884 | 0.1570 | 22.65 | 16.07 | 36.88 | 63.18 | 73.94 | $-0.98_{\pm0.56}$ |
| IMTL | 53.63 | 75.44 | 0.3868 | 0.1592 | 22.58 | 15.85 | 37.44 | 63.52 | 74.09 | $-0.57_{\pm0.24}$ |
| DB-MTL (**ours**) | 53.92 | 75.60 | **0.3768** | 0.1557 | **21.97** | 15.37 | 38.43 | 64.81 | **75.24** | $+1.15_{\pm0.16}$ |

## 4.1 SCENE UNDERSTANDING

**Datasets.** The following datasets are used: (i) *NYUv2* (Silberman et al., 2012), which is an indoor scene understanding dataset. It has 3 tasks (13-class semantic segmentation, depth estimation, and surface normal prediction) with 795 training and 654 testing images. (ii) *Cityscapes* (Cordts et al., 2016), which is an urban scene understanding dataset. It has 2 tasks (7-class semantic segmentation and depth estimation) with 2,975 training and 500 testing images.

**Baselines.** The proposed DB-MTL is compared with various types of MTL baselines, including (i) equal weighting (EW) (Zhang & Yang, 2022); (ii) GLS (Chennupati et al., 2019), which min-imizes the geometric mean loss $\sqrt[T]{\prod_{t=1}^{T} \ell_t(\mathcal{D}_t; \boldsymbol{\theta}, \boldsymbol{\psi}_t)}$; (iii) RLW (Lin et al., 2022), in which the task weights are sampled from the standard normal distribution; (iv) *loss balancing* methods including UW (Kendall et al., 2018), DWA (Liu et al., 2019a), IMTL-L (Liu et al., 2021b), and IGBv2 (Dai et al., 2023); (v) *gradient balancing* methods including MGDA (Sener & Koltun, 2018), GradNorm (Chen et al., 2018b), PCGrad (Yu et al., 2020), GradDrop (Chen et al., 2020), GradVac (Wang et al., 2021), IMTL-G (Liu et al., 2021b), CAGrad (Liu et al., 2021a), MTAdam (Malkiel & Wolf, 2021), Nash-MTL (Navon et al., 2022), MetaBalance (He et al., 2022), MoCo (Fernando et al., 2023), and Aligned-MTL (Senushkin et al., 2023); (vi) *hybrid balancing* method that com-bines loss and gradient balancing: IMTL (Liu et al., 2021b). For comparison, we also include the *single-task learning* (STL) baseline, which learns each task separately.

All methods are implemented based on the open-source `LibMTL` library (Lin & Zhang, 2023). For all MTL methods, the hard-parameter sharing (*HPS*) pattern (Caruana, 1993) is used, which consists of a task-sharing feature encoder and $T$ task-specific heads.

**Performance Evaluation.** Following Lin et al. (2022); Liu et al. (2019a), we use (i) the mean intersection over union (mIoU) and class-wise pixel accuracy (PAcc) for semantic segmentation; (ii) relative error (RErr) and absolute error (AErr) for depth estimation; (iii) mean and median an-gle errors, and percentage of normals within $t°$ ($t = 11.25, 22.5, 30$) for surface normal prediction. Following Lin et al. (2022); Maninis et al. (2019); Vandenhende et al. (2021), we report the relative performance improvement of an MTL method $\mathcal{A}$ over STL, averaged over all the metrics above, i.e.,

$$\Delta_{\mathrm{p}}(\mathcal{A}) = \frac{1}{T} \sum_{t=1}^{T} \Delta_{\mathrm{p},t}(\mathcal{A}), \quad \text{where } \Delta_{\mathrm{p},t}(\mathcal{A}) = 100\% \times \frac{1}{N_t} \sum_{i=1}^{N_t} (-1)^{s_{t,i}} \frac{M_{t,i}^{\mathcal{A}} - M_{t,i}^{\mathrm{STL}}}{M_{t,i}^{\mathrm{STL}}}. \quad (2)$$

Table 2: Performance on *Cityscapes* with 2 tasks. $\uparrow$ ($\downarrow$) indicates that the higher (lower) the result, the better the performance. The best and second best results are highlighted in **bold** and underline, respectively.

| | Segmentation | | Depth Estimation | | $\Delta_p\uparrow$ |
|---|---|---|---|---|---|
| | mIoU$\uparrow$ | PAcc$\uparrow$ | AErr$\downarrow$ | RErr$\downarrow$ | |
| STL | 69.06 | 91.54 | 0.01282 | 43.53 | 0.00 |
| EW | 68.93 | 91.58 | 0.01315 | 45.90 | $-2.05_{\pm0.56}$ |
| GLS | 68.69 | 91.45 | 0.01280 | 44.13 | $-0.39_{\pm1.06}$ |
| RLW | 69.03 | 91.57 | 0.01343 | 44.77 | $-1.91_{\pm0.21}$ |
| UW | 69.03 | 91.61 | 0.01338 | 45.89 | $-2.45_{\pm0.68}$ |
| DWA | 68.97 | 91.58 | 0.01350 | 45.10 | $-2.24_{\pm0.28}$ |
| IMTL-L | 68.98 | 91.59 | 0.01340 | 45.32 | $-2.15_{\pm0.88}$ |
| IGBv2 | 68.44 | 91.31 | 0.01290 | 45.03 | $-1.31_{\pm0.61}$ |
| MGDA | 69.05 | 91.53 | 0.01280 | 44.07 | $-0.19_{\pm0.30}$ |
| GradNorm | 68.97 | 91.60 | 0.01320 | 44.88 | $-1.55_{\pm0.70}$ |
| PCGrad | 68.95 | 91.58 | 0.01342 | 45.54 | $-2.36_{\pm1.17}$ |
| GradDrop | 68.85 | 91.54 | 0.01354 | 44.49 | $-2.02_{\pm0.74}$ |
| GradVac | 68.98 | 91.58 | 0.01322 | 46.43 | $-2.45_{\pm0.54}$ |
| IMTL-G | 69.04 | 91.54 | 0.01280 | 44.30 | $-0.46_{\pm0.67}$ |
| CAGrad | 68.95 | 91.60 | 0.01281 | 45.04 | $-0.87_{\pm0.88}$ |
| MTAdam | 68.43 | 91.26 | 0.01340 | 45.62 | $-2.74_{\pm0.20}$ |
| Nash-MTL | 68.88 | 91.52 | **0.01265** | 45.92 | $-1.11_{\pm0.21}$ |
| MetaBalance | 69.02 | 91.56 | 0.01270 | 45.91 | $-1.18_{\pm0.58}$ |
| MoCo | **69.62** | **91.76** | 0.01360 | 45.50 | $-2.40_{\pm1.50}$ |
| Aligned-MTL | 69.00 | 91.59 | 0.01270 | 44.54 | $-0.43_{\pm0.44}$ |
| IMTL | 69.07 | 91.55 | 0.01280 | 44.06 | $-0.32_{\pm0.10}$ |
| DB-MTL (**ours**) | 69.17 | 91.56 | 0.01280 | **43.46** | $+0.20_{\pm0.40}$ |

where $T$ is the number of tasks, $N_t$ is the number of metrics for task $t$, $M_{t,i}^{\mathcal{A}}$ is the $i$th metric value of method $\mathcal{A}$ on task $t$, and $s_{t,i}$ is 0 if a larger value indicates better performance for the $i$th metric on task $t$, and 1 otherwise. Each experiment is repeated three times. Further implementation details can be found in Appendix B.

**Results.** Table 1 shows the results on *NYUv2*. As can be seen, the proposed DB-MTL performs the best in terms of $\Delta_p$. Note that most of MTL baselines perform better than STL on semantic segmentation and depth estimation tasks but has a large drop on surface normal prediction task, due to the task balancing problem. Only the proposed DB-MTL has comparable performance with STL on surface normal prediction task and maintains the superiority on the other tasks, which demonstrates its effectiveness.

Table 2 shows the results on *Cityscapes*. As can be seen, DB-MTL again achieves the best in terms of $\Delta_p$. Note that all MTL baselines perform worse than STL in terms of $\Delta_p$ and only the proposed DB-MTL outperforms STL on all tasks.

## 4.2 IMAGE CLASSIFICATION

**Datasets.** The following datasets are used: (i) *Office-31* (Saenko et al., 2010), which contains $4,110$ images from three domains (tasks): Amazon, DSLR, and Webcam. Each task has 31 classes. (ii) *Office-Home* (Venkateswara et al., 2017), which contains $15,500$ images from four domains (tasks): artistic images, clipart, product images, and real-world images. Each task has 65 object categories collected under office and home settings. We use the commonly-used data split as in Lin et al. (2022): 60% for training, 20% for validation, and 20% for testing.

**Results.** Tables 3 and 4 show the results on *Office-31* and *Office-Home*, respectively, using the same set of baselines as in Section 4.1. The testing accuracy of each task is reported and the average testing accuracy among tasks and $\Delta_p$ in Eq. (2) are used as the overall performance metrics. On *Office-31*, DB-MTL achieves the top testing accuracy on DSLR and Webcam tasks over all baselines and comparable performance on Amazon task. On *Office-Home*, the performance of DB-MTL on the Artistic, Product, and Real tasks are top two. On both datasets, DB-MTL achieves the best average testing accuracy and $\Delta_p$, showing its effectiveness and demonstrating balancing both loss scale and gradient magnitude is effective.

Table 3: Classification accuracy (%) on *Office-31* with 3 tasks. ↑ indicates that the higher the result, the better the performance. The best and second best results are highlighted in **bold** and underline, respectively. Results of MoCo are from Fernando et al. (2023).

| | **Amazon** | **DSLR** | **Webcam** | **Avg**↑ | $\Delta_{\mathrm{p}}$↑ |
|---|---|---|---|---|---|
| STL | **86.61** | 95.63 | 96.85 | 93.03 | 0.00 |
| EW | 83.53 | 97.27 | 96.85 | $92.55_{\pm 0.62}$ | $-0.61_{\pm 0.67}$ |
| GLS | 82.84 | 95.62 | 96.29 | $91.59_{\pm 0.58}$ | $-1.63_{\pm 0.61}$ |
| RLW | 83.82 | 96.99 | 96.85 | $92.55_{\pm 0.89}$ | $-0.59_{\pm 0.95}$ |
| UW | 83.82 | 97.27 | 96.67 | $92.58_{\pm 0.84}$ | $-0.56_{\pm 0.90}$ |
| DWA | 83.87 | 96.99 | 96.48 | $92.45_{\pm 0.56}$ | $-0.70_{\pm 0.62}$ |
| IMTL-L | 84.04 | 96.99 | 96.48 | $92.50_{\pm 0.52}$ | $-0.63_{\pm 0.58}$ |
| IGBv2 | 84.52 | 98.36 | 98.05 | 93.64$_{\pm 0.26}$ | $+0.56_{\pm 0.25}$ |
| MGDA | 85.47 | 95.90 | 97.03 | $92.80_{\pm 0.14}$ | $-0.27_{\pm 0.15}$ |
| GradNorm | 83.58 | 97.26 | 96.85 | $92.56_{\pm 0.87}$ | $-0.59_{\pm 0.94}$ |
| PCGrad | 83.59 | 96.99 | 96.85 | $92.48_{\pm 0.53}$ | $-0.68_{\pm 0.57}$ |
| GradDrop | 84.33 | 96.99 | 96.30 | $92.54_{\pm 0.42}$ | $-0.59_{\pm 0.46}$ |
| GradVac | 83.76 | 97.27 | 96.67 | $92.57_{\pm 0.73}$ | $-0.58_{\pm 0.78}$ |
| IMTL-G | 83.41 | 96.72 | 96.48 | $92.20_{\pm 0.89}$ | $-0.97_{\pm 0.95}$ |
| CAGrad | 83.65 | 95.63 | 96.85 | $92.04_{\pm 0.79}$ | $-1.14_{\pm 0.85}$ |
| MTAdam | 85.52 | 95.62 | 96.29 | $92.48_{\pm 0.87}$ | $-0.60_{\pm 0.93}$ |
| Nash-MTL | 85.01 | 97.54 | 97.41 | $93.32_{\pm 0.82}$ | $+0.24_{\pm 0.89}$ |
| MetaBalance | 84.21 | 95.90 | 97.40 | $92.50_{\pm 0.28}$ | $-0.63_{\pm 0.30}$ |
| MoCo | 84.33 | 97.54 | 98.33 | 93.39 | - |
| Aligned-MTL | 83.36 | 96.45 | 97.04 | $92.28_{\pm 0.46}$ | $-0.90_{\pm 0.48}$ |
| IMTL | 83.70 | 96.44 | 96.29 | $92.14_{\pm 0.85}$ | $-1.02_{\pm 0.92}$ |
| DB-MTL (**ours**) | 85.12 | **98.63** | **98.51** | **94.09**$_{\pm 0.19}$ | **+1.05**$_{\pm 0.20}$ |

Table 4: Classification accuracy (%) on *Office-Home* with 4 tasks. ↑ indicates that the higher the result, the better the performance. The best and second best results are highlighted in **bold** and underline, respectively. Results of MoCo are from Fernando et al. (2023).

| | **Artistic** | **Clipart** | **Product** | **Real** | **Avg**↑ | $\Delta_{\mathrm{p}}$↑ |
|---|---|---|---|---|---|---|
| STL | 65.59 | **79.60** | **90.47** | 80.00 | 78.91 | 0.00 |
| EW | 65.34 | 78.04 | 89.80 | 79.50 | $78.17_{\pm 0.37}$ | $-0.92_{\pm 0.59}$ |
| GLS | 64.51 | 76.85 | 89.83 | 79.56 | $77.69_{\pm 0.27}$ | $-1.58_{\pm 0.46}$ |
| RLW | 64.96 | 78.19 | 89.48 | **80.11** | $78.18_{\pm 0.12}$ | $-0.92_{\pm 0.14}$ |
| UW | 65.97 | 77.65 | 89.41 | 79.28 | $78.08_{\pm 0.30}$ | $-0.98_{\pm 0.46}$ |
| DWA | 65.27 | 77.64 | 89.05 | 79.56 | $77.88_{\pm 0.28}$ | $-1.26_{\pm 0.49}$ |
| IMTL-L | 65.90 | 77.28 | 89.37 | 79.38 | $77.98_{\pm 0.38}$ | $-1.10_{\pm 0.61}$ |
| IGBv2 | 65.59 | 77.57 | 89.79 | 78.73 | $77.92_{\pm 0.21}$ | $-1.21_{\pm 0.22}$ |
| MGDA | 64.19 | 77.60 | 89.58 | 79.31 | $77.67_{\pm 0.20}$ | $-1.61_{\pm 0.34}$ |
| GradNorm | 66.28 | 77.86 | 88.66 | 79.60 | $78.10_{\pm 0.63}$ | $-0.90_{\pm 0.93}$ |
| PCGrad | 66.35 | 77.18 | 88.95 | 79.50 | $77.99_{\pm 0.19}$ | $-1.04_{\pm 0.32}$ |
| GradDrop | 63.57 | 77.86 | 89.23 | 79.35 | $77.50_{\pm 0.23}$ | $-1.86_{\pm 0.24}$ |
| GradVac | 65.21 | 77.43 | 89.23 | 78.95 | $77.71_{\pm 0.19}$ | $-1.49_{\pm 0.28}$ |
| IMTL-G | 64.70 | 77.17 | 89.61 | 79.45 | $77.98_{\pm 0.38}$ | $-1.10_{\pm 0.61}$ |
| CAGrad | 64.01 | 77.50 | 89.65 | 79.53 | $77.73_{\pm 0.16}$ | $-1.50_{\pm 0.29}$ |
| MTAdam | 62.23 | 77.86 | 88.73 | 77.94 | $76.69_{\pm 0.65}$ | $-2.94_{\pm 0.85}$ |
| Nash-MTL | 66.29 | 78.76 | 90.04 | **80.11** | 78.80$_{\pm 0.52}$ | −0.08$_{\pm 0.69}$ |
| MetaBalance | 64.01 | 77.50 | 89.72 | 79.24 | $77.61_{\pm 0.42}$ | $-1.70_{\pm 0.54}$ |
| MoCo | 63.38 | 79.41 | 90.25 | 78.70 | 77.93 | - |
| Aligned-MTL | 64.33 | 76.96 | 89.87 | 79.93 | $77.77_{\pm 0.70}$ | $-1.50_{\pm 0.89}$ |
| IMTL | 64.07 | 76.85 | 89.65 | 79.81 | $77.59_{\pm 0.29}$ | $-1.72_{\pm 0.45}$ |
| DB-MTL (**ours**) | **67.42** | 77.89 | 90.43 | 80.07 | **78.95**$_{\pm 0.35}$ | **+0.17**$_{\pm 0.44}$ |

## 4.3 MOLECULAR PROPERTY PREDICTION

**Dataset.** The following dataset is used: *QM9* (Ramakrishnan et al., 2014), which is a molecular property prediction dataset with 11 tasks. Each task performs regression on one property. We use the commonly-used split as in Fey & Lenssen (2019); Navon et al. (2022): 110, 000 for training, 10, 000 for validation, and 10, 000 for testing. Following Fey & Lenssen (2019); Navon et al. (2022), we use the mean absolute error (MAE) for performance evaluation on each task.

**Results.** Table 5 shows each task's testing MAE and overall performance $\Delta_{\mathrm{p}}$ (Eq. (2)) on *QM9*, using the same set of baselines as in Section 4.1. Note that *QM9* is a challenging dataset in MTL

Table 5: Performance on *QM9* with 11 tasks. ↑ (↓) indicates that the higher (lower) the result, the better the performance. The best and second best results are highlighted in **bold** and underline, respectively.

| | $\mu$ | $\alpha$ | $\epsilon_{\text{HOMO}}$ | $\epsilon_{\text{LUMO}}$ | $\langle R^2 \rangle$ | ZPVE | $U_0$ | $U$ | $H$ | $G$ | $c_v$ | $\Delta_{\text{p}}$↑ |
|---|---|---|---|---|---|---|---|---|---|---|---|---|
| STL | **0.062** | **0.192** | **58.82** | **51.95** | **0.529** | 4.52 | 63.69 | 60.83 | 68.33 | 60.31 | **0.069** | **0.00** |
| EW | 0.096 | 0.286 | 67.46 | 82.80 | 4.655 | 12.4 | 128.3 | 128.8 | 129.2 | 125.6 | 0.116 | $-146.3_{\pm7.86}$ |
| GLS | 0.332 | 0.340 | 143.1 | 131.5 | 1.023 | **4.45** | **53.35** | **53.79** | **53.78** | **53.34** | 0.111 | $-81.16_{\pm15.5}$ |
| RLW | 0.112 | 0.331 | 74.59 | 90.48 | 6.015 | 15.6 | 156.0 | 156.8 | 157.3 | 151.6 | 0.133 | $-200.9_{\pm13.4}$ |
| UW | 0.336 | 0.382 | 155.1 | 144.3 | 0.965 | 4.58 | 61.41 | 61.79 | 61.83 | 61.40 | 0.116 | $-92.35_{\pm13.9}$ |
| DWA | 0.103 | 0.311 | 71.55 | 87.21 | 4.954 | 13.1 | 134.9 | 135.8 | 136.3 | 132.0 | 0.121 | $-160.9_{\pm16.7}$ |
| IMTL-L | 0.277 | 0.355 | 150.1 | 135.2 | 0.946 | 4.46 | 58.08 | 58.43 | 58.46 | 58.06 | 0.110 | $-77.06_{\pm11.1}$ |
| IGBv2 | 0.235 | 0.377 | 132.3 | 139.9 | 2.214 | 5.90 | 64.55 | 65.06 | 65.12 | 64.28 | 0.121 | $-99.86_{\pm10.4}$ |
| MGDA | 0.181 | 0.325 | 118.6 | 92.45 | 2.411 | 5.55 | 103.7 | 104.2 | 104.4 | 103.7 | 0.110 | $-103.0_{\pm8.62}$ |
| GradNorm | 0.114 | 0.341 | 67.17 | 84.66 | 7.079 | 14.6 | 173.2 | 173.8 | 174.4 | 168.9 | 0.147 | $-227.5_{\pm1.85}$ |
| PCGrad | 0.104 | 0.293 | 75.29 | 88.99 | 3.695 | 8.67 | 115.6 | 116.0 | 116.2 | 113.8 | 0.109 | $-117.8_{\pm3.97}$ |
| GradDrop | 0.114 | 0.349 | 75.94 | 94.62 | 5.315 | 15.8 | 155.2 | 156.1 | 156.6 | 151.9 | 0.136 | $-191.4_{\pm9.62}$ |
| GradVac | 0.100 | 0.299 | 68.94 | 84.14 | 4.833 | 12.5 | 127.3 | 127.8 | 128.1 | 124.7 | 0.117 | $-150.7_{\pm7.41}$ |
| IMTL-G | 0.670 | 0.978 | 220.7 | 249.7 | 19.48 | 55.6 | 1109 | 1117 | 1123 | 1043 | 0.392 | $-1250_{\pm90.9}$ |
| CAGrad | 0.107 | 0.296 | 75.43 | 88.59 | 2.944 | 6.12 | 93.09 | 93.68 | 93.85 | 92.32 | 0.106 | $-87.25_{\pm1.51}$ |
| MTAdam | 0.593 | 1.352 | 232.3 | 419.0 | 24.31 | 69.7 | 1060 | 1067 | 1070 | 1007 | 0.627 | $-1403_{\pm203}$ |
| Nash-MTL | 0.115 | 0.263 | 85.54 | 86.62 | 2.549 | 5.85 | 83.49 | 83.88 | 84.05 | 82.96 | 0.097 | $-73.92_{\pm2.12}$ |
| MetaBalance | 0.090 | 0.277 | 70.50 | 78.43 | 4.192 | 11.2 | 113.7 | 114.2 | 114.5 | 111.7 | 0.110 | $-125.1_{\pm7.98}$ |
| MoCo | 0.489 | 1.096 | 189.5 | 247.3 | 34.33 | 64.5 | 754.6 | 760.1 | 761.6 | 720.3 | 0.522 | $-1314_{\pm65.2}$ |
| Aligned-MTL | 0.123 | 0.295 | 98.07 | 94.56 | 2.397 | 5.90 | 86.42 | 87.42 | 87.19 | 86.75 | 0.106 | $-80.58_{\pm4.18}$ |
| IMTL | 0.138 | 0.344 | 106.1 | 102.9 | 2.595 | 7.84 | 102.5 | 103.0 | 103.2 | 100.8 | 0.110 | $-104.3_{\pm11.7}$ |
| DB-MTL (**ours**) | 0.112 | 0.264 | 89.26 | 86.59 | 2.429 | 5.41 | 60.33 | 60.78 | 60.80 | 60.59 | 0.098 | $-58.10_{\pm3.89}$ |

and none of the MTL methods performs better than STL, as observed in previous works (Gasteiger et al., 2020; Navon et al., 2022). DB-MTL performs the best among all MTL methods and greatly improves over the second-best MTL method, Nash-MTL, in terms of $\Delta_{\text{p}}$.

### 4.4 ABLATION STUDY

DB-MTL has two components: the loss-scale balancing method (i.e., logarithm transformation) in Section 3.1 and the gradient-magnitude balancing method in Section 3.2. In this experiment, we study the effectiveness of each component. We consider the four combinations: (i) use neither loss-scale nor gradient-magnitude balancing methods (i.e., the EW baseline); (ii) use only loss-scale balancing; (iii) use only gradient-magnitude balancing; (iv) use both loss-scale and gradient-magnitude balancing methods (i.e., the proposed DB-MTL). Table 6 shows the $\Delta_{\text{p}}$ of different combinations on five datasets, i.e., *NYUv2*, *Cityscapes*, *Office-31*, *Office-Home*, and *QM9*. As can be seen, on all datasets, both components are beneficial to DB-MTL and combining them achieves the best performance.

Table 6: Ablation study of DB-MTL on different datasets in terms of $\Delta_{\text{p}}$.

| loss-scale balancing | gradient-magnitude balancing | *NYUv2* | *Cityscapes* | *Office-31* | *Office-Home* | *QM9* |
|---|---|---|---|---|---|---|
| ✗ | ✗ | $-1.78_{\pm0.45}$ | $-2.05_{\pm0.56}$ | $-0.61_{\pm0.67}$ | $-0.92_{\pm0.59}$ | $-146.3_{\pm7.86}$ |
| ✓ | ✗ | $+0.06_{\pm0.09}$ | $-0.38_{\pm0.39}$ | $+0.93_{\pm0.42}$ | $-0.73_{\pm0.95}$ | $-74.40_{\pm13.2}$ |
| ✗ | ✓ | $+0.76_{\pm0.25}$ | $+0.12_{\pm0.70}$ | $+0.01_{\pm0.39}$ | $-0.78_{\pm0.49}$ | $-65.73_{\pm2.86}$ |
| ✓ | ✓ | $\mathbf{+1.15_{\pm0.16}}$ | $\mathbf{+0.20_{\pm0.40}}$ | $\mathbf{+1.05_{\pm0.20}}$ | $\mathbf{+0.17_{\pm0.44}}$ | $\mathbf{-58.10_{\pm3.89}}$ |

## 5 CONCLUSION

In this paper, we alleviate the task-balancing problem in MTL by presenting Dual-Balancing Multi-Task Learning (DB-MTL), a novel approach that consists of loss-scale and gradient-magnitude balancing methods. The former ensures all task losses have the same scale via the logarithm transformation, while the latter guarantees that all task gradients have the same magnitude as the maximum gradient norm by a gradient normalization. Extensive experiments on a number of benchmark datasets demonstrate that DB-MTL achieves state-of-the-art performance and the logarithm transformation can benefit existing gradient balancing methods.

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

# APPENDIX

## A ANALYSIS

**Proposition A.1.** *For $x > 0$, $\log(x) = \min_s e^s x - s - 1$.*

*Proof.* Define an auxiliary function $f(s) = e^s x - s - 1$. It is easy to show that $\frac{\mathrm{d}f(s)}{\mathrm{d}s} = e^s x - 1$ and $\frac{\mathrm{d}^2 f(s)}{\mathrm{d}s^2} = e^s x > 0$. Thus, $f(s)$ is convex. By the first-order optimal condition (Boyd & Vandenberghe, 2004), let $e^{s^\star} x - 1 = 0$, the global minimizer is solved as $s^\star = -\log(x)$. Therefore, $f(s^\star) = e^{s^\star} x - s^\star - 1 = e^{-\log(x)} x + \log(x) - 1 = \log(x)$, where we finish the proof. $\square$

## B IMPLEMENTATION DETAILS FOR SECTION 4

***NYUv2*** **and** ***Cityscapes***. Following Lin et al. (2022), we use the *DeepLabV3+* network (Chen et al., 2018a), containing a *ResNet-50* network with dilated convolutions pre-trained on the *ImageNet* dataset (Deng et al., 2009) as the shared encoder and the *Atrous Spatial Pyramid Pooling* (Chen et al., 2018a) module as the task-specific head. We train the model for 200 epochs by using the Adam optimizer (Kingma & Ba, 2015) with learning rate $10^{-4}$ and weight decay $10^{-5}$. The learning rate is halved to $5 \times 10^{-5}$ after 100 epochs. The cross-entropy loss, $L_1$ loss, and cosine loss are used as the loss functions of the semantic segmentation, depth estimation, and surface normal prediction tasks, respectively. All input images are resized to $288 \times 384$, and the batch size is set to 8 for training for *NYUv2*. We resize the input images to $128 \times 256$, and use the batch size 64 for training for *Cityscapes*.

***Office-31*** **and** ***Office-Home***. Following Lin et al. (2022), a *ResNet-18* (He et al., 2016) pre-trained on the *ImageNet* dataset (Deng et al., 2009) is used as a shared encoder, and a linear layer is used as a task-specific head. We resize the input image to $224 \times 224$. The batch size and the training epoch are set to 64 and 100, respectively. The Adam optimizer (Kingma & Ba, 2015) with the learning rate as $10^{-4}$ and the weight decay as $10^{-5}$ is used. The cross-entropy loss is used for all the tasks and classification accuracy is used as the evaluation metric.

***QM9***. Following Fey & Lenssen (2019); Navon et al. (2022), a graph neural network (Gilmer et al., 2017) is used as the shared encoder, and a linear layer is used as the task-specific head. The targets of each task are normalized to have zero mean and unit standard deviation. The batch size and training epoch are set to 128 and 300, respectively. The Adam optimizer (Kingma & Ba, 2015) with the learning rate 0.001 is used for training and the ReduceLROnPlateau scheduler (Paszke et al., 2019) is used to reduce the learning rate once $\Delta_{\mathrm{p}}$ on the validation dataset stops improving. Following Fey & Lenssen (2019); Navon et al. (2022), we use mean squared error (MSE) as the loss function.

Following Fernando et al. (2023), for each dataset, we perform grid search for $\beta$ over $\{0.1, 0.5, 0.9, \frac{0.1}{k^{0.5}}, \frac{0.5}{k^{0.5}}, \frac{0.9}{k^{0.5}}\}$, where $k$ is the number of iterations.

## C ADDITIONAL EXPERIMENTAL RESULTS

### C.1 EFFECTS OF MTL ARCHITECTURES

The proposed DB-MTL is agnostic to the choice of MTL architectures. In this section, we evaluate DB-MTL on *NYUv2* using two more MTL architectures: *Cross-stitch* (Misra et al., 2016) and *MTAN* (Liu et al., 2019a). Two current state-of-the-art GLS and CAGrad are compared with. The implementation details are the same as those in *HPS* architecture in Appendix B. Figure 5 shows each task's improvement performance $\Delta_{\mathrm{p},t}$. For *Cross-stitch* (Figure 5(a)), as can be seen, DB-MTL performs the best on all tasks, showing its effectiveness. As for *MTAN* (Figure 5(b)), compared with STL, the MTL methods (GLS, CAGrad, and DB-MTL) perform better on both semantic segmentation and depth estimation tasks, but only DB-MTL achieves comparable performance on the surface normal prediction task.

Furthermore, we conduct experiments to evaluate DB-MTL on *NYUv2* with *SegNet* network (Badrinarayanan et al., 2017). The implementation details are same as those in Appendix B, except that

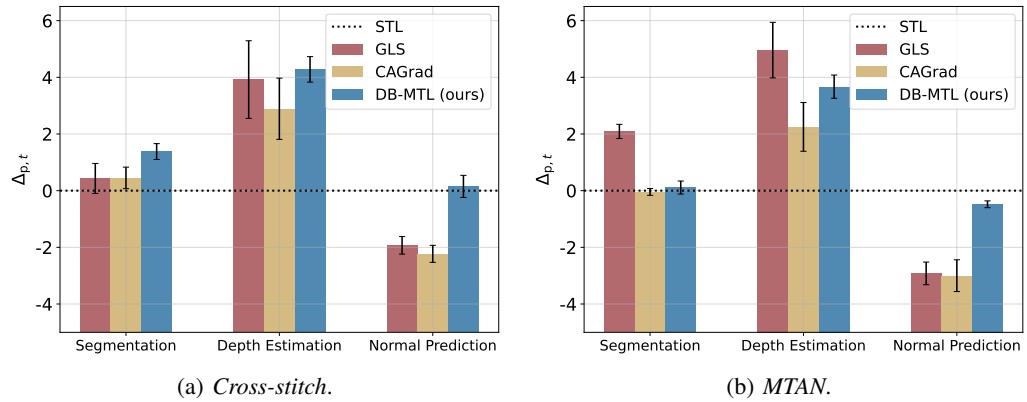

(a) *Cross-stitch*.  (b) *MTAN*.

Figure 5: Performance on *NYUv2* for *Cross-stitch* and *MTAN* architectures.

the batch size is set to 2 and the data augmentation is used following Liu et al. (2021a). The results are shown in Table 7. As can be seen, DB-MTL again achieves the best performance in terms of $\Delta_{\mathrm{p}}$, demonstrating the effectiveness of the proposed method.

Table 7: Performance on the *NYUv2* dataset with *SegNet* network. $\uparrow$ ($\downarrow$) indicates that the higher (lower) the result, the better the performance. The best and second best results are highlighted in **bold** and underline, respectively. Superscripts $\sharp$, $\S$, $\ddagger$, and $*$ denote the results are from Liu et al. (2021a), Navon et al. (2022), Fernando et al. (2023), and Senushkin et al. (2023), respectively.

| | Segmentation | | Depth Estimation | | Surface Normal Prediction | | | | | |
| --- | --- | --- | --- | --- | --- | --- | --- | --- | --- | --- |
| | | | | | Angle Distance | | Within $t°$ | | | |
| | mIoU↑ | PAcc↑ | AErr↓ | RErr↓ | Mean↓ | MED↓ | 11.25↑ | 22.5↑ | 30↑ | $\Delta_{\mathrm{p}}$↑ |
| STL$^\S$ | 38.30 | 63.76 | 0.6754 | 0.2780 | 25.01 | 19.21 | 30.14 | 57.20 | 69.15 | 0.00 |
| EW$^\S$ | 39.29 | 65.33 | 0.5493 | 0.2263 | 28.15 | 23.96 | 22.09 | 47.50 | 61.08 | +0.88 |
| GLS | 39.78 | 65.63 | 0.5318 | 0.2272 | 26.13 | 21.08 | 26.57 | 52.83 | 65.78 | +5.15 |
| RLW$^\S$ | 37.17 | 63.77 | 0.5759 | 0.2410 | 28.27 | 24.18 | 22.26 | 47.05 | 60.62 | −2.16 |
| UW$^\S$ | 36.87 | 63.17 | 0.5446 | 0.2260 | 27.04 | 22.61 | 23.54 | 49.05 | 63.65 | +0.91 |
| DWA$^\S$ | 39.11 | 65.31 | 0.5510 | 0.2285 | 27.61 | 23.18 | 24.17 | 50.18 | 62.39 | +1.93 |
| IMTL-L | 39.78 | 65.27 | 0.5408 | 0.2347 | 26.26 | 20.99 | 26.42 | 53.03 | 65.94 | +4.39 |
| IGBv2 | 38.03 | 64.29 | 0.5489 | 0.2301 | 26.94 | 22.04 | 24.77 | 50.91 | 64.12 | +2.11 |
| MGDA$^\S$ | 30.47 | 59.90 | 0.6070 | 0.2555 | 24.88 | 19.45 | 29.18 | 56.88 | 69.36 | −1.66 |
| GradNorm$^*$ | 20.09 | 52.06 | 0.7200 | 0.2800 | **24.83** | **18.86** | **30.81** | **57.94** | **69.73** | −11.7 |
| PCGrad$^\S$ | 38.06 | 64.64 | 0.5550 | 0.2325 | 27.41 | 22.80 | 23.86 | 49.83 | 63.14 | +1.11 |
| GradDrop$^\S$ | 39.39 | 65.12 | 0.5455 | 0.2279 | 27.48 | 22.96 | 23.38 | 49.44 | 62.87 | +2.07 |
| GradVac$^*$ | 37.53 | 64.35 | 0.5600 | 0.2400 | 27.66 | 23.38 | 22.83 | 48.66 | 62.21 | −0.49 |
| IMTL-G$^\S$ | 39.35 | 65.60 | 0.5426 | 0.2256 | 26.02 | 21.19 | 26.20 | 53.13 | 66.24 | +4.77 |
| CAGrad$^\sharp$ | 39.18 | 64.97 | 0.5379 | 0.2229 | 25.42 | 20.47 | 27.37 | 54.73 | 67.73 | +5.81 |
| MTAdam | 39.44 | 65.73 | 0.5326 | 0.2211 | 27.53 | 22.70 | 24.04 | 49.61 | 62.69 | +3.21 |
| Nash-MTL$^\S$ | 40.13 | 65.93 | 0.5261 | 0.2171 | 25.26 | 20.08 | 28.40 | 55.47 | 68.15 | +7.65 |
| MetaBalance | 39.85 | 65.13 | 0.5445 | 0.2261 | 27.35 | 22.66 | 23.70 | 49.69 | 63.09 | +2.67 |
| MoCo$^\ddagger$ | 40.30 | 66.07 | 0.5575 | **0.2135** | 26.67 | 21.83 | 25.61 | 51.78 | 64.85 | +4.85 |
| Aligned-MTL$^*$ | 40.82 | 66.33 | 0.5300 | 0.2200 | 25.19 | 19.71 | 28.88 | 56.23 | 68.54 | +8.16 |
| IMTL | 41.19 | 66.37 | 0.5323 | 0.2237 | 26.06 | 20.77 | 26.76 | 53.48 | 66.32 | +6.45 |
| DB-MTL (ours) | **41.42** | **66.45** | **0.5251** | 0.2160 | 25.03 | 19.50 | 28.72 | 56.17 | 68.73 | **+8.91** |

## C.2 Effects of $\alpha_k$

Figure 6 shows the results of different strategies of $\alpha_k$ on the *Cityscapes*, *Office-31*, *Office-Home*, and *QM9* datasets, respectively. As can be seen, the strategy of normalizing to the maximum gradient norm is consistently better.

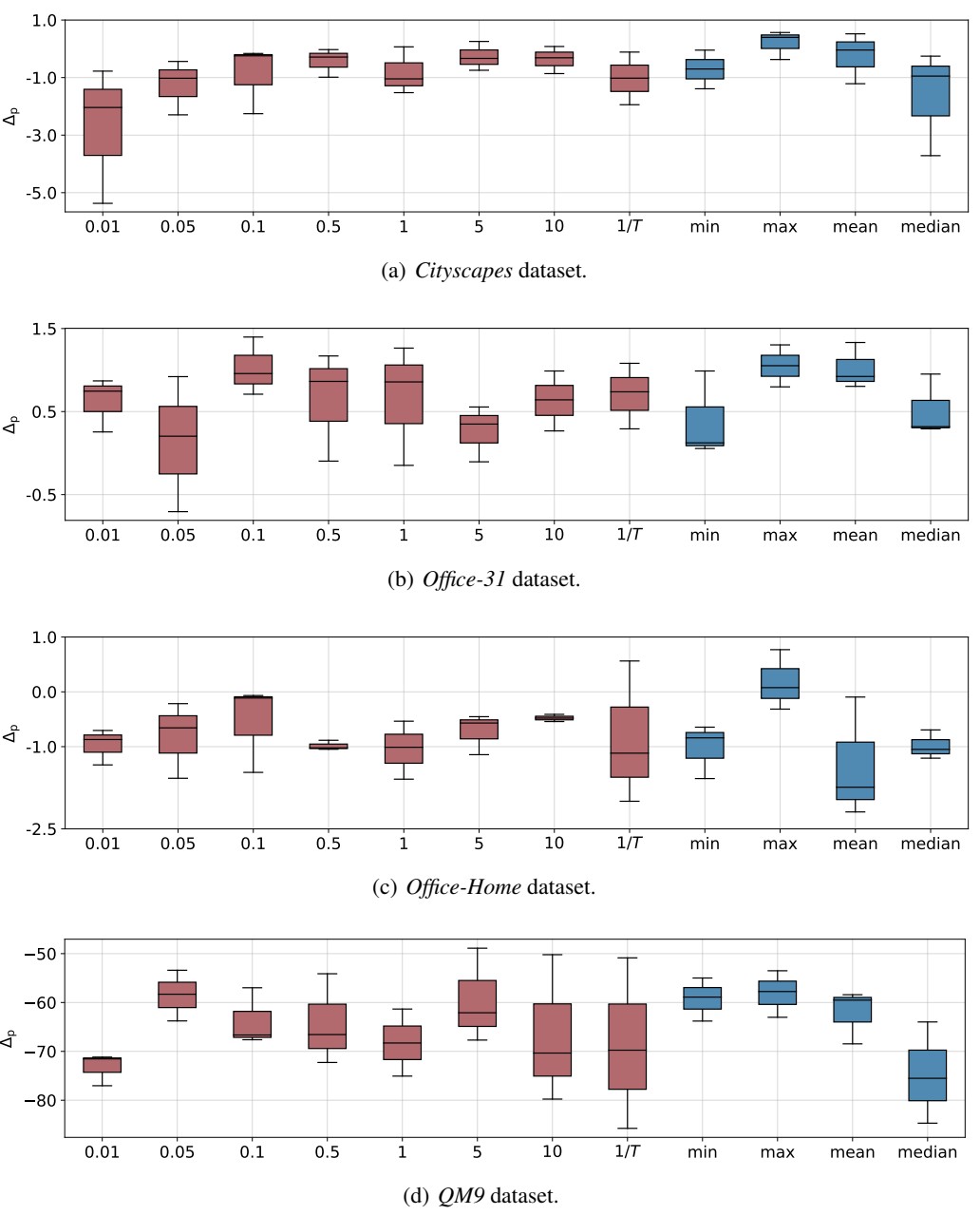

(a) *Cityscapes* dataset.

(b) *Office-31* dataset.

(c) *Office-Home* dataset.

(d) *QM9* dataset.

Figure 6: Results of different strategies for $\alpha_k$ in Eq. (1) on different datasets in terms of $\Delta_{\mathrm{p}}$. "min", "max", "mean", and "median" denotes the minimum, maximum, average, and median of $\|\hat{\boldsymbol{g}}_{t,k}\|_2$ $(t = 1, \ldots, T)$, respectively.

## C.3 Effects of $\beta$

Table 8 shows the results of DB-MTL with different $\beta$ on *Office-31* dataset in terms of the average classification accuracy and $\Delta_{\mathrm{p}}$. As can be seen, DB-MTL is insensitive over a large range of $\beta$ (i.e., $\{0.1/k^{0.5}, 0.2/k^{0.5}, ..., 0.9/k^{0.5}\}$) and performs better than DB-MTL without EMA ($\beta = 0$).

Table 8: Effects of $\beta$ on *Office-31* dataset.

| $\beta$ | **Avg↑** | $\mathbf{\Delta_p}$↑ |
|---|---|---|
| 0 | $93.89_{\pm 0.09}$ | $+0.84_{\pm 0.09}$ |
| $0.1/k^{0.5}$ | $94.09_{\pm 0.19}$ | $+1.05_{\pm 0.20}$ |
| $0.2/k^{0.5}$ | $94.02_{\pm 0.42}$ | $+0.97_{\pm 0.47}$ |
| $0.3/k^{0.5}$ | $94.15_{\pm 0.51}$ | $+1.12_{\pm 0.55}$ |
| $0.4/k^{0.5}$ | $94.20_{\pm 0.22}$ | $+1.18_{\pm 0.23}$ |
| $0.5/k^{0.5}$ | $94.01_{\pm 0.56}$ | $+0.97_{\pm 0.60}$ |
| $0.6/k^{0.5}$ | $94.05_{\pm 0.56}$ | $+1.01_{\pm 0.61}$ |
| $0.7/k^{0.5}$ | $94.01_{\pm 0.26}$ | $+0.96_{\pm 0.30}$ |
| $0.8/k^{0.5}$ | $94.00_{\pm 0.16}$ | $+0.94_{\pm 0.19}$ |
| $0.9/k^{0.5}$ | $94.09_{\pm 0.08}$ | $+1.07_{\pm 0.08}$ |
| 0.1 | $93.79_{\pm 0.29}$ | $+0.73_{\pm 0.29}$ |
| 0.2 | $93.15_{\pm 0.61}$ | $+0.06_{\pm 0.67}$ |
| 0.3 | $92.91_{\pm 0.41}$ | $-0.23_{\pm 0.45}$ |
| 0.4 | $93.08_{\pm 0.68}$ | $-0.01_{\pm 0.70}$ |
| 0.5 | $92.47_{\pm 0.69}$ | $-0.68_{\pm 0.74}$ |
| 0.6 | $92.41_{\pm 0.31}$ | $-0.76_{\pm 0.35}$ |
| 0.7 | $92.52_{\pm 0.36}$ | $-0.69_{\pm 0.36}$ |
| 0.8 | $92.29_{\pm 0.28}$ | $-0.94_{\pm 0.34}$ |
| 0.9 | $90.67_{\pm 0.56}$ | $-2.74_{\pm 0.66}$ |

## C.4 Analysis of Training Efficiency

Figure 7 shows the per-epoch running time of different MTL methods on *NYUv2* dataset. All methods are tested over 100 epochs on an NVIDIA GeForce RTX 3090 GPU and the average running time per epoch is reported. As can be seen, DB-MTL has a similar running time to gradient balancing methods and IMTL, but it is larger than loss balancing methods because each task's gradient is computed in every iteration (i.e., step 6 in Algorithm 1). It is a common problem in gradient balancing methods (Liu et al., 2021a; Navon et al., 2022; Sener & Koltun, 2018; Yu et al., 2020). Although DB-MTL is slower than loss balancing methods, it achieves better performance, as shown in Tables 1, 2, 3, 4, and 5.

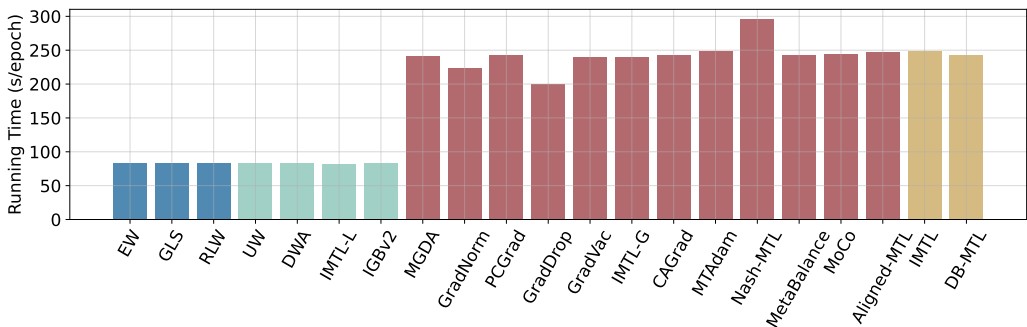

Figure 7: The running time per epoch averaged over 100 repetitions of different methods on *NYUv2* dataset.

### C.5 ANALYSIS OF TRAINING STABILITY

Figure 8 shows each task's training loss and gradient norm of EW and DB-MTL on *Office-31* dataset. As can be seen, for the proposed DB-MTL, both training loss and gradient norm for every task decrease smoothly and finally converge, which means both the logarithm transformation and the maximum-norm strategy do not affect the training stability. Moreover, DB-MTL converges faster than EW because of balancing on both loss and gradient levels.

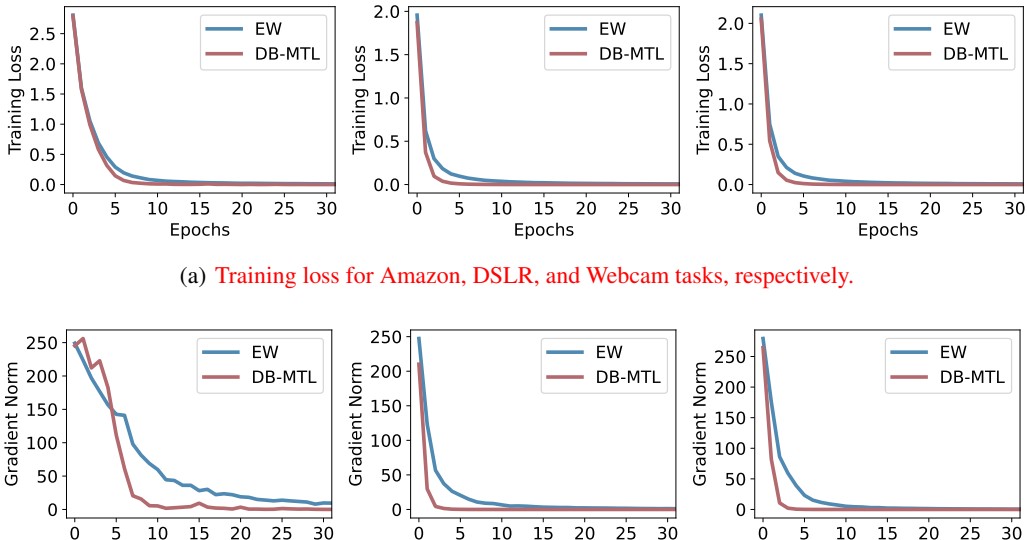

(a) Training loss for Amazon, DSLR, and Webcam tasks, respectively.

(b) Gradient norm for Amazon, DSLR, and Webcam tasks, respectively.

Figure 8: Each task's training loss and gradient norm $\mathbb{E}_{\mathcal{B}_{t,k}} \|\nabla_{\boldsymbol{\theta}_k} \ell_t(\mathcal{B}_{t,k}; \boldsymbol{\theta}_k, \boldsymbol{\psi}_{t,k})\|^2$ of EW and DB-MTL on *Office-31* dataset.

## D DISCUSSION WITH AUXILIARY LEARNING

Similar to multi-task learning, auxiliary learning (Du et al., 2019; Liu et al., 2022; Navon et al., 2021) improves the learning of tasks with the help of some auxiliary tasks. Hence, the task balancing problem also exists in auxiliary learning. However, the goals of multi-task learning and auxiliary learning are different: the former expects all tasks to perform well but the latter only focuses on the main task(s). Hence, it is challenging to apply the proposed DB-MTL to auxiliary learning directly. We will study it in the future.

