# OpenReview forum: "Dual-Balancing for Multi-Task Learning"
_ICLR.cc/2024/Conference — Submitted to ICLR 2024_

### Official Review · Reviewer_8dJR · 2023-10-28

**Soundness:** 3 good
**Presentation:** 3 good
**Contribution:** 3 good
**Rating:** 8
**Confidence:** 4

**Summary:**

The paper tackles the problem of multi-task learning by proposing a dual-balancing strategy in the hard shared paradigm. For the task-shared parameters, gradients from different tasks are normalized to the maximum norm. For the task-specific parameters, logarithm transformations are applied to the loss so that all losses have similar scales. Ablation studies show that each of the proposed strategies can enhance other complementary methods. Besides, combining the two balancing can surpass all previous methods.

**Strengths:**

1. The paper proposes two novel strategies in the context of hard parameter sharing multi-task learning, one for gradient balancing and the other for loss balancing, which are well-motivated and easy to follow.
2. For each of the two strategies, it can boost other complementary methods when integrated, showing the general applicability.
3. The authors conduct extensive experiments on scene understanding, image classification and molecular property prediction to show the effectiveness of the dual-balancing strategy.

**Weaknesses:**

1. It would be better to compare the training time efficiency of each method.
2. More discussions can be made on the proposed method, such that the stability, e.g. will the balancing method cause the model to collapse during training?

**Questions:**

The authors could provide the training dynamics (both loss and gradient) such that readers can better understand how the balancing strategy affects the model learning.

---

> ### Author Response · Authors · 2023-11-20
> **Reply to Reviewer 8dJR**
>
> Thanks for your thoughtful review and valuable feedback. We address your concerns as follows.
>
> ---
>
> > Q1: It would be better to compare the training time efficiency of each method.
>
> **A1**: As suggested, we provided the comparison of training efficiency for all methods in Figure 7 of the updated paper (Appendix C.4). We can see that the proposed DB-MTL is as efficient as existing gradient balancing methods since all task gradients are computed in every iteration (Line 6 in Algorithm 1 of the paper).
>
> ---
>
> > Q2: More discussions can be made on the proposed method, such that the stability, e.g. will the balancing method cause the model to collapse during training?
>
> **A2**: As suggested, we conducted an empirical analysis on the training stability of DB-MTL. Figure 8 (in Appendix C.5 of the updated paper) shows the curves of training loss and gradient norm for EW and DB-MTL on tasks of the *Office-31* dataset. As can be seen, DB-MTL converges as smoothly as EW, suggesting the proposed balancing methods do not affect the training stability.
>
> ---
>
> > Q3: The authors could provide the training dynamics (both loss and gradient) such that readers can better understand how the balancing strategy affects the model learning.
>
> **A3**: We provided the training dynamics, including the training loss and gradient norm during the training process in Appendix C.5 of the updated paper. We can see that DB-MTL, which balances tasks across both loss and gradient levels, converges faster than EW.

---

> ### Comment · Reviewer_8dJR · 2023-11-21
>
> Thank the authors for the rebuttal. I would keep my original rating.

---

> > ### Author Response · Authors · 2023-11-21
> >
> > Thanks again for your positive comments.

---

### Official Review · Reviewer_SooC · 2023-10-30

**Soundness:** 3 good
**Presentation:** 3 good
**Contribution:** 3 good
**Rating:** 6
**Confidence:** 5

**Summary:**

This paper proposes a fairly simple but effective multi-task optimisation strategy with two key components: i) Optimising on log loss for scale balancing; and ii) applying l2-normalisation on aggregated gradients via EMA for gradient balancing. Each component can actually be combined with existing multi-task optimisation strategies, and have shown improved performance to validate the effectiveness. The paper presents experiments in standard multi-task learning benchmarks on dense prediction tasks, image classification tasks and molecular property prediction tasks.

**Strengths:**

1.	Idea is clean and simple and can be universally applied in many multi-task learning optimisation strategies.
2.	Experiments are well-rounded and covered in many settings and domains.
3.	Ablation shows the importance of each design component, presenting gradient-magnitude-balancing possibly is more important than loss-scale balancing.

**Weaknesses:**

No prominent weaknesses. See questions.

**Questions:**

1.	Since each design component shows to be effective and useful for existing multi-task optimisation strategy, I think it’s important to highlight DB could be considered as a drop-in enhancement to existing multi-task methods to further improve performance.
2.	From the algorithm 1, it looks like the method should be able to run very fast, as similar to standard weight-based methods. Maybe it’s also useful to highlight its running speed, comparing to other gradient-based methods such as CAGrad, Nash-MTL which take considerably longer training time.
3.	It might be also interesting to see whether DB could also enhance auxiliary learning methods, such as AuxiLearn (Navon et al 2021), GCS (Du et al 2018) and Auto-Lambda (Liu et al 2022). This could further emphasise the design benefits and may approach even wider range of audiences.

---

> ### Author Response · Authors · 2023-11-20
> **Reply to Reviewer SooC**
>
> Thanks for your thoughtful review and valuable feedback. We address your concerns as follows.
>
> ---
>
> > Q1: Since each design component shows to be effective and useful for existing multi-task optimisation strategy, I think it’s important to highlight DB could be considered as a drop-in enhancement to existing multi-task methods to further improve performance.
>
> **A1**: Thanks for your suggestions. We revised the paper accordingly to highlight the logarithm transformation is beneficial to existing gradient balancing methods in the contribution and conclusion parts.
>
> ---
>
> > Q2: From the algorithm 1, it looks like the method should be able to run very fast, as similar to standard weight-based methods. Maybe it’s also useful to highlight its running speed, comparing to other gradient-based methods such as CAGrad, Nash-MTL which take considerably longer training time.
>
> **A2**: As suggested, we added the running time for all methods in Figure 7 of the update paper (Appendix C.4). We can see that the proposed DB-MTL runs as fast as previous gradient balancing methods since all task gradients are required to compute at every iteration (Line 6 in Algorithm 1 of the paper).
>
> ---
>
> > Q3: It might be also interesting to see whether DB could also enhance auxiliary learning methods, such as AuxiLearn (Navon et al 2021), GCS (Du et al 2018) and Auto-Lambda (Liu et al 2022). This could further emphasise the design benefits and may approach even wider range of audiences.
>
> **A3**: The goals of multi-task learning and auxiliary learning are different: the former expects all tasks to perform well but the latter only focuses on the main task(s). Hence, it is challenging to apply our balancing methods to auxiliary learning directly. We added discussions on the mentioned papers in Appendix D.

---

> > ### Comment · Reviewer_SooC · 2023-11-21
> > **Response**
> >
> > Thanks for the response and additional evaluations. I believe these additional analyses have resolved all my concerns. I have no other questions and would love to maintain my original rating as weak accept.

---

> > > ### Author Response · Authors · 2023-11-22
> > >
> > > Thanks for your reply. We are glad that our reply addressed your concerns.

---

### Official Review · Reviewer_gC4S · 2023-11-01

**Soundness:** 2 fair
**Presentation:** 2 fair
**Contribution:** 2 fair
**Rating:** 3
**Confidence:** 4

**Summary:**

This paper proposes a dual-balancing multi-task learning (DB-MTL) method to alleviate the task-balancing problem in multi-task learning (MTL). DB-MLT attempts to combine a loss-scale balancing and a gradient-magnitude balancing. Specifically, the former performs a logarithm transformation on each task loss to ensure all losses have the same scale. The latter normalizes the gradients of all tasks to have the same magnitude as the maximum gradient norm. This guarantees the gradients contribute equally. Combining the two methods, DB-MTL achieves state-of-the-art performance on several MTL benchmark datasets, including scene understanding, image classification, and molecular property prediction. Ablation studies demonstrate both the loss-scale and gradient-magnitude balancing components are effective.

**Strengths:**

- The method is well-motivated. Experiments on various datasets demonstrate state-of-the-art performance. Ablations validate the efficacy of each component.
- The dual-balancing framework is reasonable. The loss-scale balancing via logarithmic transformation helps existing gradient methods. Normalizing gradients by the maximum norm is simple yet effective.
- The problem setup and methods are clearly explained.

**Weaknesses:**

- The novelty of this paper is limited. Loss and gradient balance have been extensively explored, and this paper does not have any new technical insights. It is only a combination of the existing works.
- The theoretical analysis is not sufficient to suppor the advantage of the proposed method. Proving convergence guarantees or other theoretical properties of DB-MTL could reveal more advantages over existing methods.
- Detailed experimental analyzing on why DB-MTL outperforms certain baselines is missing.
- The discription of the intuition is not clear.

**Questions:**

- Can you provide any theoretical analysis for DB-MTL? Convergence guarantees or other properties could reveal advantages over existing methods.
- For the loss transformation, is there an adaptive way to estimate the scale? Using the maximum seems to work well, but a learned loss scale could be interesting.

---

> ### Author Response · Authors · 2023-11-20
> **Reply to Reviewer gC4S (1/2)**
>
> Thanks for your thoughtful review and valuable feedback. We address your concerns as follows.
>
> ----
>
> > Q1: The novelty of this paper is limited. Loss and gradient balance have been extensively explored, and this paper does not have any new technical insights. It is only a combination of the existing works.
>
> **A1:** This paper aims to alleviate the task balancing problem from loss and gradient perspectives. The proposed method is simple yet effective and largely outperforms baselines on five datasets.
>
> (i) As discussed in Section 3.1 of the submission,
> logarithm transformation is only used as a baseline studied in Nash-MTL and its advantages for MTL methods are **NOT** researched there. In our paper, we conducted extensive experiments to show that logarithm transformation is beneficial to various MTL methods. As shown in Figure 1 of the submission, MTL methods integrated with logarithm transformation can boost performance by a large margin. The effectiveness of combining logarithm transformation with MTL methods, **one of our major contributions**, is useful for the MTL community.
>
> (ii) As discussed in Section 3.2 of the submission, the proposed gradient balancing method is different from GradNorm: firstly, GradNorm cannot guarantee all task gradients to have the same magnitude in each iteration since it updates the model parameters and task weights alternately; secondly, GradNorm does not consider the effect of the update magnitude (i.e., $\alpha_k$ in our paper), which significantly affects the performance (as shown in Figures 4 and 6 in our paper).
>
> We are the **first** to propose normalizing task gradients to the same magnitude as the maximum gradient norm, and extensively experimental results (like Figure 3, Tables 1 and 6 in the submission) show it achieves better performance than GradNorm and other gradient balancing methods, which is **one of our major contributions**.
>
> (iii) Logarithm transformation and the proposed gradient normalization are two **simple but effective** techniques, which are complementary and can be simply combined together as the proposed DB-MTL. Compared with IMTL, which is complex in learning weights for balancing both losses and gradients, DB-MTL is simpler and more effective (Tables 1, 2, 3, 4, and 5 in the submission). Furthermore, by a simple combination, DB-MTL achieves state-of-the-art performance on various datasets, which is our primary contribution.
>
> (iv) Recent works [1, 2] have conducted extensive experiments to demonstrate the most simple method, equal weighting (EW in our paper), can perform comparably or even better than many complex multi-task/multi-objective methods via tuning some hyperparameters (like the learning rate) in [1] or using some regularization and stabilization techniques in [2]. Hence, we believe proposing simple but effective methods for the MTL field is useful.
>
> ---
>
> > Q2: The theoretical analysis is not sufficient to support the advantage of the proposed method. Proving convergence guarantees or other theoretical properties of DB-MTL could reveal more advantages over existing methods.
>
> > Can you provide any theoretical analysis for DB-MTL? Convergence guarantees or other properties could reveal advantages over existing methods.
>
>
> **A2**: As mentioned in our reply to Q1, recent works [1, 2] have conducted extensive experiments to demonstrate the performance of the existing MTL methods are unsatisfactory, although some of them have convergence guarantees. Besides, as discussed in MoCo [3], many MTL methods (like MGDA, PCGrad, CAGrad, etc.) provide the convergence analysis in the deterministic case but they are stochastic algorithms in practice. Although MoCo has a convergence guarantee in the practice stochastic setting, its performance is worse than the proposed DB-MTL (results are reported in Tables 1-5 in the submission). Hence, we believe that it is more important to propose an effective method for the current MTL field. In our paper, extensive experiments on five datasets consistently show that the proposed DB-MTL achieves better performance than existing methods. We will conduct the challenging theoretical analysis in the future.

---

> ### Author Response · Authors · 2023-11-20
> **Reply to Reviewer gC4S (2/2)**
>
> > Q3: Detailed experimental analyzing on why DB-MTL outperforms certain baselines is missing.
>
> **A3**: In the submission, we have conducted extensive experiments on various datasets and provided analysis to demonstrate the superiority of DB-MTL over a large number of baselines. As shown in Tables 1-5, DB-MTL achieves state-of-the-art performance on all five datasets. This is attributed to
>
> (i) Logarithm transformation is a simple but effective loss balancing method. As shown in Figure 1 in the submission, integrating logarithm transformation into various gradient balancing methods boosts the performance largely. Moreover, logarithm transformation is more effective than IMTL-L (Figure 2 in the submission).
>
> (ii) Our gradient-magnitude balancing method is more effective than the existing gradient normalization method (i.e., GradNorm), as shown in Figure 3 of the submission.
>
> (iii) The ablation study in Section 4.4 demonstrates the effectiveness of using loss-scale balancing, gradient-magnitude balancing, and combining them together.
>
> ---
>
> > Q4: The discription of the intuition is not clear.
>
> **A4**: We suppose the reviewer asks clarification for the intuition of the choice of $\alpha_k$. We further clarify it as follows.
>
> (i) When some tasks have large gradient norms and others have small gradient norms, the model $\theta_k$ is close to a point where the former tasks have not yet converged while the latter tasks have converged. This point is unsatisfactory in MTL, as we hope all tasks achieve convergence. If we use the min-norm strategy, the model will be caught by such a point. Hence, a large $\alpha_k$ is more suitable to escape the unsatisfactory point, and the max-norm strategy is adopted.
>
> (ii) When all task gradient norms are small, the model $\theta_k$ is close to a stationary point of all tasks (which is a satisfactory solution in MTL). Even using the max-norm strategy, the $\alpha_k$ is small as all task gradient norms are small. Hence, the model does NOT escape the stationary point.
>
> ---
>
> > Q5: For the loss transformation, is there an adaptive way to estimate the scale? Using the maximum seems to work well, but a learned loss scale could be interesting.
>
> **A5**: As discussed in Section 3.1 of the submission, previous works (e.g., IMTL-L) attempt to learn loss scales by **approximately** solving an optimization problem by one-step gradient descent at every iteration. Our Proposition A.1 in Appendix shows the logarithm transformation is equivalent to IMTL-L when the optimization problem is solved **exactly**. Hence, the loss transformation is better than IMTL-L in solving an optimization problem of learning loss scales. Experimental results (Figure 2 in the submission) show that logarithm transformation is more effective than IMTL-L.
>
> ---
>
> **References**
>
> [1] Kurin et al. In Defense of the Unitary Scalarization for Deep Multi-Task Learning. In *Neural Information Processing Systems*, 2022.
>
> [2] Xin et al. Do Current Multi-Task Optimization Methods in Deep Learning even Help? In *Neural Information Processing Systems*, 2022.
>
> [3] Fernando et al. Mitigating Gradient Bias in Multi-objective Learning: A Provably Convergent Approach. In *International Conference on Learning Representations*, 2023.

---

> ### Author Response · Authors · 2023-11-22
> **Have our reply addressed your concerns?**
>
> Dear Reviewer gC4S,
>
> Thank you again for your detailed reviews. We have responded to your comments in the above reply and we hope that our reply has satisfactorily addressed your concerns.
>
> If there is any additional explanation or experiments that can save the reviewer’s time to understand our paper and clarify the concerns, we will be more than happy to do so.
>
> Best,
>
> The authors

---

### Official Review · Reviewer_H1EY · 2023-11-02

**Soundness:** 2 fair
**Presentation:** 2 fair
**Contribution:** 2 fair
**Rating:** 5
**Confidence:** 4

**Summary:**

The paper presents a straightforward Dual-Balancing Multi-Task Learning (DB-MTL) method with well-conducted experiments across multiple datasets. The method alleviate the task balancing problem by using logarithmic transformation for loss-scale balancing and normalizing gradients to the same magnitude for gradient-magnitude balancing.

**Strengths:**

1. **Simplicity of the Method:** The loss-scale balancing and the gradient-magnitude balancing of the approach are both commendably straightforward.
2. **Sufficient and Effective Experiments:** The paper demonstrates the effectiveness of DB-MTL through extensive validation across three distinct scenarios and five datasets. The proposed method is optimal on all datasets.

**Weaknesses:**

1. **Imprecise Overview of the MTL Objective:** The MTL objective (Equation 1) in Section 2 is imprecise. It is applicable primarily to existing loss balancing and certain gradient balancing methods. For example, in some gradient balancing methods like PCGrad and CAGrad, the weights of task-specific parameters are all 1, and in all hybrid balancing methods, the weights of task-specific and task-shared parameters are different.
2. **Omission of Relevant Literature:**   (1) The DB-MTL method, albeit not defined as such within the text, is one of hybrid balancing methods. However, the treatment of hybrid balancing in Section 2 is overly simplistic and lacks a formal definition, along with an inadequate reference to relevant literature. For instance, approaches like RLW, Auto-λ[1] and IGB[2] have also conducted exploratory work combining loss balancing and gradient balancing.    (2) In the discussion part of Section 3.1, IGB[2] has also studied the logarithm transformation and has further extended it into a new loss balancing paradigm.
3. **Lack of Novelty:** The proposed loss balancing method, as well as the combination of loss and gradient balancing, have been previously discussed and experimented in prior works.
4. **Lack of Theoretical Analysis:** Though the proposed method is easy to understand intuitively, it lacks theoretical understanding about why the proposed gradient balancing is beneficial for the overall MTL objective. In terms of gradients, the primary factor affecting the MTL performance is gradient conflicts, specifically the conflicts in gradient directions. The discrepancy in the magnitude of gradients does not lead to performance degradation if their directions are aligned. How does the proposed gradient balancing method directly or indirectly mitigate the conflicts in gradient directions? Are there any theoretical analyses, empirical experiments (maybe toy examples), or even intuitive explanations?
5. **Fairness of Comparative Experiments:** Combining loss balancing and gradient balancing is beneficial. Thus, directly comparing MB-MTL with existing gradient balancing methods might introduce unfairness. In the image classification scenario, when only the proposed gradient-magnitude balancing is employed, the performance is not optimal within gradient balancing methods (NashMTL yields better results). Would combining logarithmic transformation with NashMTL surpass MB-MTL?

[1] Auto-λ: Disentangling Dynamic Task Relationships. TMLR, 2022.\
[2] Improvable Gap Balancing for Multi-Task Learning. UAI, 2023.

**Questions:**

See my comments in Weaknesses.

---

> ### Author Response · Authors · 2023-11-20
> **Reply to Reviewer H1EY (1/2)**
>
> Thank you for your thoughtful review and valuable feedback. We address your concerns as follows.
>
> ---
>
> > Q1: Imprecise Overview of the MTL Objective: The MTL objective (Equation 1) in Section 2 is imprecise. It is applicable primarily to existing loss balancing and certain gradient balancing methods. For example, in some gradient balancing methods like PCGrad and CAGrad, the weights of task-specific parameters are all 1, and in all hybrid balancing methods, the weights of task-specific and task-shared parameters are different.
>
> **A1**: We have revised Section 2 for clarification:
>
> (i) In the "Loss Balancing Methods" paragraph: For loss balancing methods, the task weight $\gamma_t$ affects both the update of both task-sharing parameter $\theta$ and task-specific parameter $\psi$.
>
> (ii) In the "Gradient Balancing Methods" paragraph: For most gradient balancing methods (e.g., PCGrad, CAGrad, MoCo, GradDrop, and IMTL), the task weight $\gamma_t$ affects the update of task-sharing parameter $\theta$ only. In some gradient balancing methods (e.g., GradNorm, MGDA, and Nash-MTL), task weight $\gamma_t$ plays the same role as in loss balancing methods.
>
> (iii) In the "Hybrid Balancing Methods" paragraph: For hybrid balancing methods, the task weight $\gamma_t$ is the product of loss and gradient balancing weights.
>
> ---
>
> > Q2: Omission of Relevant Literature: (1) The DB-MTL method, albeit not defined as such within the text, is one of hybrid balancing methods. However, the treatment of hybrid balancing in Section 2 is overly simplistic and lacks a formal definition, along with an inadequate reference to relevant literature. For instance, approaches like RLW, Auto-$\lambda$[1] and IGB[2] have also conducted exploratory work combining loss balancing and gradient balancing. (2) In the discussion part of Section 3.1, IGB[2] has also studied the logarithm transformation and has further extended it into a new loss balancing paradigm.
>
> **A2**: (i) Combining loss balancing and gradient balancing is first proposed in IMTL (as discussed in Related Work in the submission), while the mentioned three follow-up works (RLW, Auto-$\lambda$ [1], and IGB [2]) combine their balancing methods with some existing loss/gradient balancing methods. We have added the discussion in Section 2 of the updated paper.
>
> (ii) We have revised Section 3.1 accordingly by adding a discussion with IGB. (1) IGB combines the logarithm transformation with **loss** balancing methods to achieve better performance. In our paper, we demonstrate that logarithm transformation benefits existing **gradient** balancing methods, which is a contribution complementary to the exploration in the IGB paper. (2) Furthermore, IGB does not clearly explain why the logarithm transformation can work well. We show that it can address the loss scale problem in multi-task learning.
>
> (iii) We have added IGBv2 as a baseline and compared it with our DB-MTL on five benchmark datasets. As shown in Tables 1-5 of the updated paper, DB-MTL consistently outperforms IGBv2.
>
> ---
>
> > Q3: Lack of Novelty: The proposed loss balancing method, as well as the combination of loss and gradient balancing, have been previously discussed and experimented in prior works.
>
> **A3**: (i) The logarithm transformation is one of the components of our proposed DB-MTL, but we do not claim it is proposed in this paper. As discussed in Section 3.1 of the submission, logarithm transformation is only used as a baseline studied in Nash-MTL. Although IGB combines the logarithm transformation with loss balancing methods to achieve better performance, IGB also does not clearly explain why the logarithm transformation can work well. In our paper, we show that the logarithm transformation can address the loss scale problem in multi-task learning, which has not been studied in previous works. Besides, we conduct extensive experiments to show that logarithm transformation is beneficial to existing gradient balancing methods, which is a contribution complementary to the exploration in the IGB paper. As shown in Figure 1 of the submission, existing gradient balancing methods integrated with logarithm transformation can boost performance by a large margin. The effectiveness of combining logarithm transformation with gradient balancing methods, **one of our major contributions**, is useful for the MTL community.
>
> (ii) Logarithm transformation and the proposed gradient normalization are two **simple but effective** techniques, which are complementary and can be simply combined together as the proposed DB-MTL. Compared with IMTL, which is complex in learning weights for balancing both losses and gradients, DB-MTL is simpler and more effective (Tables 1, 2, 3, 4, and 5 in the submission). Furthermore, by a simple combination, DB-MTL achieves state-of-the-art performance on various datasets, which is our primary contribution.

---

> ### Author Response · Authors · 2023-11-20
> **Reply to Reviewer H1EY (2/2)**
>
> > Q4: Lack of Theoretical Analysis: Though the proposed method is easy to understand intuitively, it lacks theoretical understanding about why the proposed gradient balancing is beneficial for the overall MTL objective. In terms of gradients, the primary factor affecting the MTL performance is gradient conflicts, specifically the conflicts in gradient directions. The discrepancy in the magnitude of gradients does not lead to performance degradation if their directions are aligned. How does the proposed gradient balancing method directly or indirectly mitigate the conflicts in gradient directions? Are there any theoretical analyses, empirical experiments (maybe toy examples), or even intuitive explanations?
>
> **A4**: (i) Recent works [3, 4] have conducted extensive experiments to demonstrate the performance of the existing MTL methods are unsatisfactory. Hence, we believe that it is more important to propose an effective method for the current MTL field. In our paper, extensive experiments on five datasets consistently show that the proposed DB-MTL achieves better performance than existing methods. We will conduct the challenging theoretical analysis in the future.
>
> (ii) Intuitively, both gradient-magnitude discrepancies and gradient-direction conflicts can affect the MTL performance, but which one is more crucial is open. A very recent work [5] observes that the negative transfer and gradient-direction conflicts are not strongly correlated. Our experimental results show that the proposed gradient balancing method (i.e., simply normalizing all gradients as the maximum-norm one) is more effective than previous complex gradient manipulating methods like eliminating gradient-direction conflicts in PCGrad. Thus, it seems that gradient-direction conflicts are not crucial to mitigating negative transfer in MTL. This problem is still open.
>
> \begin{array}{lccccc}\hline& \text{\textit{NYUv2}} & \text{\textit{Cityscapes}} & \text{\textit{Office-31}} & \text{\textit{Office-Home}} & \text{\textit{QM9}} \newline\hline\text{PCGrad} & -1.57_{\pm0.44} & -2.36_{\pm1.17} & -0.68_{\pm0.57} & -1.04_{\pm0.32} & -117.8_{\pm3.97} \newline\text{GradVac} & -1.75_{\pm0.39} & -2.45_{\pm0.54} & -0.58_{\pm0.78} & -1.49_{\pm0.28} & -150.7_{\pm7.41} \newline\text{gradient-magnitude balancing (\textbf{ours})} & \mathbf{+0.76}\_{\pm0.25} & \mathbf{+0.12}\_{\pm0.70} & \mathbf{+0.01}\_{\pm0.39} & \mathbf{-0.78}\_{\pm0.49} & \mathbf{-65.73}\_{\pm2.86} \newline\hline\end{array}
>
> ---
>
> > Q5: Fairness of Comparative Experiments: Combining loss balancing and gradient balancing is beneficial. Thus, directly comparing MB-MTL with existing gradient balancing methods might introduce unfairness. In the image classification scenario, when only the proposed gradient-magnitude balancing is employed, the performance is not optimal within gradient balancing methods (NashMTL yields better results). Would combining logarithmic transformation with NashMTL surpass MB-MTL?
>
> **A5**: As suggested, we conducted additional experiments on image classification scenarios (*Office-31* and *Office-Home*) to compare "Nash-MTL + logarithm transformation" with DB-MTL. The table below shows the results. As can be seen, DB-MTL performs better. Moreover, integrating logarithm transformation into Nash-MTL is better than  Nash-MTL. Furthermore, on *NYUv2*, *Cityscapes*, and *QM9* datasets, the proposed gradient-magnitude balancing method (the 3rd row of Table 6 in the submission) outperforms existing gradient balancing methods (results are reported in Tables 1, 2, and 5 of the submission).
>
> \begin{array}{lcc}\hline &\text{\textit{Office-31}} & \text{\textit{Office-Home}} \newline\hline\text{Nash-MTL} & +0.24_{\pm0.89} & -0.08_{\pm0.69} \newline\text{Nash-MTL+Log} & +0.71_{\pm0.44} & +0.06_{\pm0.38} \newline\text{DB-MTL (\textbf{ours})} & \mathbf{+1.05}\_{\pm0.20} & \mathbf{+0.17}\_{\pm0.44} \newline\hline\end{array}
>
> ----
>
> **References**
>
> [1] Liu et al. Auto-Lambda: Disentangling Dynamic Task Relationships. *Transactions on Machine Learning Research*, 2022.
>
> [2] Dai et al. Improvable Gap Balancing for Multi-Task Learning. In *Uncertainty in Artificial Intelligence*, 2023.
>
> [3] Kurin et al. In Defense of the Unitary Scalarization for Deep Multi-Task Learning. In *Neural Information Processing Systems*, 2022.
>
> [4] Xin et al. Do Current Multi-Task Optimization Methods in Deep Learning even Help? In *Neural Information Processing Systems*, 2022.
>
> [5] Jiang et al. ForkMerge: Mitigating Negative Transfer in Auxiliary-Task Learning. In *Neural Information Processing Systems*, 2023.

---

> ### Author Response · Authors · 2023-11-22
> **Have our reply addressed your concerns?**
>
> Dear Reviewer H1EY,
>
> Thank you again for your detailed reviews. We have responded to your comments in the above reply and we hope that our reply has satisfactorily addressed your concerns.
>
> If there is any additional explanation or experiments that can save the reviewer’s time to understand our paper and clarify the concerns, we will be more than happy to do so.
>
> Best,
>
> The authors

---

### Official Review · Reviewer_PmP7 · 2023-11-09

**Soundness:** 2 fair
**Presentation:** 2 fair
**Contribution:** 2 fair
**Rating:** 5
**Confidence:** 4

**Summary:**

The paper presents a method to tackle the task-balancing problem in multi-task learning called Dual-Balancing Multi-Task Learning (DB-MTL). The work is grounded on the observation that differences in loss scales and gradient magnitudes across different tasks could negatively impact the performance of some tasks. The DB-MTL method is designed to balance these aspects at both the loss and gradient levels. It involves a logarithmic transformation on each task loss to ensure uniform scale, which is shown to also benefit existing gradient balancing techniques. Moreover, it also includes the normalization of all task gradients to the same magnitude to ensure respective uniformity. The researchers found that setting this magnitude as the maximum gradient norm yields the best performance. Comprehensive experiments have been performed on several benchmark datasets, and the results suggest that DB-MTL can achieve state-of-the-art performance. The primary contributions include the proposal of the DB-MTL method, empirical evidence of its effectiveness, and an illustration of the advantage of combining loss-scale balancing with gradient balancing.

**Strengths:**

Quality: The authors execute extensive experiments which support their conclusions. The postulations and the experimental results correlate well, strengthening the paper’s credibility.

Clarity: The paper is well-written and very easy to understand.

Originality: The maximum-norm strategy in Section 3.2  has some novelty.

**Weaknesses:**

There is a significant lack of novelty in the presented techniques. The first part discusses scale-balancing loss transformation, which utilizes the common method of applying a log transformation to loss. This technique has already been mentioned by Nash-MTL and therefore doesn't contribute anything substantially new to the field.
The second section of the method, gradient normalization, is a modification of the GradNorm technique. The presented maximum-norm strategy essentially ensures all small-task gradients have the same norm as the task with the greatest gradient. While this may present an interesting approach, it is not substantially innovative. The final part of the paper brings together the previous two parts, but again, this combination doesn't present any great novelty or challenges. Consequently, the paper’s primary contribution is the maximum-norm strategy, which is not significant enough.

From the writing perspective, this paper reads more like a technical report or experiment report, where effective methods from previous work are summarized, but with few unique thoughts proposed.

From an experimental perspective, the explanation of the maximum-norm strategy in section 3.2 lacks experimental support and could benefit from visual analysis. For example, the authors could investigate how using different alpha values impacts the optimization path or the decrease of loss in different tasks. This would provide a much-needed experimental grounding to the theoretical concepts presented in the paper.

Overall, the paper needs more novelty and substantial analysis to make a significant contribution to the field.

**Questions:**

Why are the baseline methods worse than STL on almost all datasets? In previous papers (IMTL etc.), at least on the NYUv2 dataset, the effects of most methods are positive.

---

> ### Author Response · Authors · 2023-11-20
> **Reply to Reviewer PmP7 (1/2)**
>
> Thanks for your thoughtful review and valuable feedback. We address your concerns as follows.
>
> ---
>
> > Q1: There is a significant lack of novelty in the presented techniques. The first part discusses scale-balancing loss transformation, which utilizes the common method of applying a log transformation to loss. This technique has already been mentioned by Nash-MTL and therefore doesn't contribute anything substantially new to the field. The second section of the method, gradient normalization, is a modification of the GradNorm technique. The presented maximum-norm strategy essentially ensures all small-task gradients have the same norm as the task with the greatest gradient. While this may present an interesting approach, it is not substantially innovative. The final part of the paper brings together the previous two parts, but again, this combination doesn't present any great novelty or challenges. Consequently, the paper’s primary contribution is the maximum-norm strategy, which is not significant enough.
>
> > From the writing perspective, this paper reads more like a technical report or experiment report, where effective methods from previous work are summarized but with few unique thoughts proposed.
>
> **A1:** This paper aims to alleviate the task balancing problem from loss and gradient perspectives. The proposed method is **simple yet effective** and largely outperforms baselines on five datasets.
>
> (i) As discussed in Section 3.1 of the submission, logarithm transformation is only used as a baseline studied in Nash-MTL and its advantages for MTL methods are **NOT** researched there. In our paper, we conducted extensive experiments to show that logarithm transformation is beneficial to various MTL methods. As shown in Figure 1 of the submission, MTL methods integrated with logarithm transformation can boost performance by a large margin. The effectiveness of combining logarithm transformation with MTL methods, **one of our major contributions**, is useful for the MTL community.
>
> (ii) As discussed in Section 3.2 of the submission, the proposed gradient balancing method is different from GradNorm: firstly, GradNorm cannot guarantee all task gradients to have the same magnitude in each iteration since it updates the model parameters and task weights alternately; secondly, GradNorm does not consider the effect of the update magnitude (i.e., $\alpha_k$ in our paper), which significantly affects the performance (as shown in Figures 4 and 6 in our paper).
>
> We are the **first** to propose normalizing task gradients to the same magnitude as the maximum gradient norm, and extensively experimental results (like Figure 3, Tables 1 and 6 in the submission) show it achieves better performance than GradNorm and other gradient balancing methods, which is **one of our major contributions**.
>
> (iii) Logarithm transformation and the proposed gradient normalization are two **simple but effective** techniques, which are complementary and can be simply combined together as the proposed DB-MTL. Compared with IMTL, which is complex in learning weights for balancing both losses and gradients, DB-MTL is simpler and more effective (Tables 1, 2, 3, 4, and 5 in the submission). Furthermore, by a simple combination, DB-MTL achieves state-of-the-art performance on various datasets, which is **our primary contribution**.
>
> (iv) Recent works [1, 2] have conducted extensive experiments to demonstrate the most simple method, equal weighting (EW in the submission), can perform comparably or even better than many complex multi-task/multi-objective methods via tuning some hyperparameters (like the learning rate) in [1] or using some regularization and stabilization techniques in [2]. Hence, we believe proposing simple but effective methods for the MTL field is useful.
>
> ---
>
>
> > Q2: From an experimental perspective, the explanation of the maximum-norm strategy in section 3.2 lacks experimental support and could benefit from visual analysis. For example, the authors could investigate how using different alpha values impacts the optimization path or the decrease of loss in different tasks. This would provide a much-needed experimental grounding to the theoretical concepts presented in the paper.
>
> **A2**: We have provided intuitive explanations and extensive empirical experiments on five datasets (Figures 4 and 6 in the submission) to demonstrate the maximum-norm strategy is reasonable. We will further study it in the future.

---

> ### Author Response · Authors · 2023-11-20
> **Reply to Reviewer PmP7 (2/2)**
>
> > Q3: Why are the baseline methods worse than STL on almost all datasets? In previous papers (IMTL etc.), at least on the NYUv2 dataset, the effects of most methods are positive.
>
> **A3**: (i) There are **two backbones SegNet and ResNet50** widely used in evaluating MTL methods on the NYUv2 dataset. The former, which is used in the mentioned work IMTL, is less powerful and MTL methods can beat STL. The submission has included the results of both backbones. **For SegNet, as can be seen from Table 7 in Appendix C.1, our DB-MTL and most of the MTL methods achieve better performance than STL, which is consistent with previous papers (e.g., IMTL).**
> For ResNet50, we can see from Table 1 in the main paper that only three MTL methods (CLS, IGBv2, and our DB-MTL) can beat STL.
>
> (ii) For Office-31 and Office-Home datasets, as observed in previous works (Table 8 in RLW paper and Tables 7 and 8 in MoCo paper), most of the baselines are worse than EW in terms of $\Delta_{\mathrm{p}}$ (EW is usually worse than STL). As discussed in the Nash-MTL paper, the QM9 dataset is challenging, and all of the existing MTL methods underperform STL.
>
> (iii) Recently, large-scale empirical studies [1, 2] also demonstrated the performance of the existing MTL methods is unsatisfactory, suggesting the importance of developing a more effective MTL algorithm.
>
> ---
>
> **References**
>
> [1] Kurin et al. In Defense of the Unitary Scalarization for Deep Multi-Task Learning. In *Neural Information Processing Systems*, 2022.
>
> [2] Xin et al. Do Current Multi-Task Optimization Methods in Deep Learning even Help? In *Neural Information Processing Systems*, 2022.

---

> ### Author Response · Authors · 2023-11-22
> **Have our reply addressed your concerns?**
>
> Dear Reviewer PmP7,
>
> Thank you again for your detailed reviews. We have responded to your comments in the above reply and we hope that our reply has satisfactorily addressed your concerns.
>
> If there is any additional explanation or experiments that can save the reviewer’s time to understand our paper and clarify the concerns, we will be more than happy to do so.
>
> Best,
>
> The authors

---

### Author Response · Authors · 2023-11-20
**Reply to all the reviewers**

Dear Reviewers,

We thank all the reviewers for their constructive and valuable comments. We have revised the paper accordingly and highlighted the changes in red. The major changes are summarized as follows.

- Adding a discussion with IGB in Section 3.1 (for Reviewer H1EY);

- Adding IGB as a baseline (for Reviewer H1EY);

- Analysis of training efficiency in Appendix C.4 (for Reviewers SooC and 8dJR);

- Analysis of training dynamic in Appendix C.5 (for Reviewer 8dJR).

We have responded to each reviewer separately. We hope that we have satisfactorily addressed your concerns. Please let us know if there are any further concerns or questions.

Best,

The authors

---

### Comment · Area_Chair_mN4L · 2023-11-23
**From AC at the end of rebuttal: Reviewer response required**

Dear Reviewers,

Thanks for your time and commitment to the ICLR 2024 review process.

As we approach the conclusion of the author-reviewer discussion period (Wednesday, Nov 22nd, AOE), I kindly urge those who haven't engaged with the authors' dedicated rebuttal to please take a moment to review their response and share your feedback, regardless of whether it alters your opinion of the paper.

Your feedback is essential to a thorough assessment of the submission.

Best regards,

AC

---

### Meta-Review · Area_Chair_mN4L · 2023-12-10

**Metareview:**

The paper presents a combined approach for multi-task learning, with the aim to balance different tasks and promote positive transfers. The paper received mixed ratings and divergent opinions, which did not converge after the rebuttal. By reading the paper and considering all reviews and author responses, it becomes clear that the technical contribution of this is below the bar. It is a combination of well-established methods, with the max-norm baring minor novelty. No technical insights are observed from either theoretical or empirical analyses. The resulting method appears as a "selection" of the pros and cons of previous methods and then combines the best ones.

**Justification For Why Not Higher Score:**

Divergent reviews with important open issues on the technical novelty and contribution.

**Justification For Why Not Lower Score:**

N/A

---

### Decision · Program_Chairs · 2024-01-16

Reject